# When Verifiable Rewards Switch the Language: Cross-lingual Collapse in Chain-of-Thought

## Abstract

Reinforcement learning with verifiable reward (RLVR) has been instrumental in eliciting strong reasoning capabilities from large language models (LLMs) via long chains of thought (CoT). During RLVR training, we formalize and systematically study an empirical phenomenon whereby a multilingual model's CoT reverts to its dominant pre-training language (e.g., English) even when prompted in another language, which we term Cross-lingual Collapse. Because the long-CoT regime magnifies exposure to linguistic priors, the underlying trade-off between maximizing reasoning depth and preserving target-language fidelity has remained under-characterized. To examine this trade-off, we train LLMs with Group-Relative Policy Optimization (GRPO) on translated versions of math datasets widely used to elicit long-CoT reasoning. Throughout training, we track both task accuracy and the language consistency of reasoning chains. Our experiments yield three findings: (i) under RLVR, CoT in LLMs systematically drifts toward the pre-training dominant language as reasoning performance rises; (ii) English-centric priors, long-CoT GRPO optimization, task difficulty, and high-entropy decoding jointly amplify this drift, and the pattern persists beyond mathematics; and (iii) interventions that favor target-language traces—via a language-consistency reward, decoding-time controls, or more balanced backbones—mitigate collapse but reveal a persistent performance–fidelity trade-off.

## 1 Introduction

Large language models (LLMs) trained with long chain-of-thought (CoT) supervision have demonstrated impressive performance across mathematically demanding problems, code generation tasks, and multi-step logical reasoning benchmarks (Wei et al., 2022; Shao et al., 2024; Yu et al., 2025; DeepSeek-AI et al., 2025). These models' strengthened reasoning capabilities not only enable human-level performance on challenging tasks but also facilitate monitoring of intermediate reasoning traces, thereby improving interpretability and enabling more reliable auditing.

Although multilingual competence has been studied during pre-training and instruction tuning (Shaham et al., 2024; Zhong et al., 2024; Kew et al., 2024; Wang et al., 2025), reasoning-centric models remain comparatively underexplored. We posit an inherent *trade-off*: pushing for deeper, verification-driven reasoning with long CoT can come at the expense of *target-language* fidelity. Mechanistically, long CoT increases exposure to pre-training priors; when those priors are English-dominant—as is the case for most open-source foundation models (OLMo et al., 2024; Grattafiori et al., 2024; Yoo et al., 2024b; Yang et al., 2025; Team et al., 2025)—reward-seeking optimization can preferentially route the reasoning trace through English even under non-English prompts. We refer to the resulting drift as **Cross-lingual Collapse**: the chain-of-thought reverts to the pre-training dominant language while task performance continues to rise.

To systematically analyze this performance–fidelity trade-off, we study target-language reasoning under reinforcement learning with verifiable reward (RLVR). We instantiate Group-Relative Policy Optimization (GRPO) (Shao et al., 2024) on an English-centric backbone (OLMo et al., 2024) and non-English-centric backbones (Grattafiori et al., 2024; Yang et al., 2025), using standard math word-problem corpora widely used to elicit long-CoT reasoning (e.g., GSM8K (Cobbe et al., 2021),

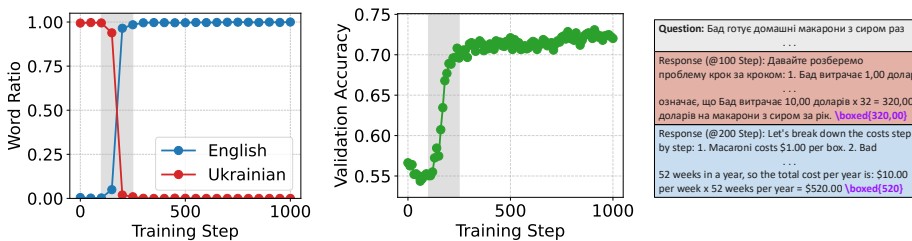

(a) Training Step vs. Word Ratio   (b) GSM8K (UK) Performance   (c) Response Examples

Figure 1: Illustration of **Cross-lingual Collapse**. We train Llama-3.2-3B Instruct with GRPO on a fully Ukrainian translation of GSM8K, seeking Ukrainian-only reasoning. **(a)** Chain-of-thought word-ratio in reward warding rollouts over training steps. In the grey band, the share of Ukrainian tokens plummets, while English abruptly dominates, signaling a language switch within the rollout reasoning trace. **(b)** Accuracy on the Ukrainian GSM8K. The sharp rise in accuracy aligns with the same 100–250-step window, showing that the model scores higher once its reasoning drifts into English. **(c)** Representative responses at steps 100 and 200 (answer spans highlighted in purple). When the model reasons in Ukrainian it produces an incorrect answer, but after switching to English it solves the problem correctly, exemplifying the collapse from target-language reasoning to the pre-training-dominant language. The word ratio is measured during training from the rollout samples.

SimpleRL-Zoo (Zeng et al., 2025)) translated into five target languages. Our evaluation tracks (i) task accuracy and (ii) a target-language word ratio over training, enabling us to quantify language drift alongside performance. Beyond measurement of Cross-lingual Collapse, we interrogate both the amplifiers and mechanisms and the mitigations and limits of this behavior. Our novelty is three-fold:

- **Phenomenon.** We formalize and operationalize Cross-lingual Collapse as a phenomenon characterized by rising accuracy with systematic drift of CoT into the pre-training dominant language, quantified via accuracy and word ratios on English and a target language.
- **Amplifiers and mechanisms.** We show that English-dominant LLMs and long-CoT GRPO optimization steer reward toward dominant-language traces, and that task difficulty and high-entropy decoding further exacerbate the drift; the pattern persists beyond math.
- **Mitigations and limits.** We evaluate intuitive interventions (language-consistency reward, decoding controls, and multilingual mixing) that partially alleviate collapse, revealing a persistent performance–fidelity trade-off rather than a one-size-fits-all solution.

## 2 MOTIVATION

Recent reinforcement learning with verifiable reward (RLVR) methods such as Group-Relative Policy Optimization (GRPO) (DeepSeek-AI et al., 2025) unlock state-of-the-art reasoning by having the model speak its thoughts aloud: each answer is preceded by a multi-step chain-of-thought that can be several hundred tokens long. With this drastic increase in utterance length, the burden on the model's linguistic competence also multiplies for every step of the trace.

In non-English contexts, this burden is even greater (Marchisio et al., 2024). For English-centric LLMs, a single error introduced during an early non-English step can propagate through the entire chain of reasoning, ultimately compromising the final answer. Early work (Shaham et al., 2024; Kew et al., 2024) demonstrated that even target-language-centric supervised fine-tuning (SFT) (Ouyang et al., 2022) on a single language can still coax a model into showing modest generalization beyond English. However, current evidence is sparse on how reasoning-driven training like GRPO affects these cross-lingual gains—do they hold steady, or do they shift?

We therefore ran a pilot experiment on the Llama-3.2-3B Instruct, giving it target-language reasoning supervision through GRPO. Concretely, we fine-tuned the model on the GSM8K grade-school arithmetic corpus, translated into Ukrainian so that all intermediate chain-of-thought steps as well as the final answer were presented in a low-resource language (relatively lower than English (Wenzek et al., 2020)). As training progressed, however, the chains gradually drifted back to high-resource

languages, chiefly English, even though the prompts remained Ukrainian. The trend is visualized in Figure 1. We dub this behavior **Cross-lingual Collapse** in reasoning models: a systematic collapse of target-lingual chains-of-thought toward the model's dominant pre-training language.

**In response,** this work aims to establish and explain Cross-lingual Collapse under RLVR: we corroborate the phenomenon across translated long-CoT settings, identify its causal drivers and triggering conditions, and examine how it can be mitigated and to what extent.

## 3 EXPERIMENTS

### 3.1 EXPERIMENTAL SETTINGS

**Base models.** To investigate the influence of foundation model design on reasoning in a target language, we categorized base models into two groups: (1) english-dominant LLMs, (2) non-English dominant LLMs. We selected OLMo2-1B Instruct as an english-dominant LLM (OLMo et al., 2024), Llama-3.2 3B Instruct (Grattafiori et al., 2024) and Qwen-2.5 1.5B Instruct(Team, 2024) as representative non-English dominant LLMs. [1] This setup allows us to investigate how the intrinsic prior of languages shape the emergence of non-English reasoning abilities when the models are prompted to reason in a variety of language.

**Training configuration.** To enhance the reasoning capability of LLMs, we train the base models with GRPO, a representative RLVR algorithm shown to strengthen reasoning. We used GSM8K training dataset, the community's most widely utilized dataset for mathematical problems (Shao et al., 2024; DeepSeek-AI et al., 2025). Training was conducted within a verl framework (Sheng et al., 2024), using a slightly modified hyperparameter configuration from the SimpleRL project Zeng et al. (2025), which are proven effective for this task. To evaluate the improvement of reasoning ability of a trained language, we translated the entire training corpus into Korean (KO), Ukrainian (UK), Chinese (ZH), Thai(TH), Japanese(JA) using GPT-4o. These languages span distinct scripts and a gradient of pre-training resource levels (high-resource: ZH/JA, mid-resource: KO, lower-resource: TH/UK) and are intended as a representative case rather than exhaustive coverage. The quality of the translated data was ensured using quality filtering (Guerreiro et al., 2024), as detailed in Appendix B.We excluded 15% of training dataset for validation. Unless otherwise noted, we train GRPO models on the translated GSM8K corpus only. In Sec. 3.3, when studying the effect of task difficulty, we utilize a mix of a 7k subset of translated SimpleRL-Zoo dataset.

**Evaluation dataset.** We evaluated our model on the translated GSM8K and MATH500 (Lightman et al., 2024) test sets across multiple languages. In order to compute the accuracy, we utilize math-verify library [2] for obtaining robust mathematical expression.

**Target Word Ratio (Target WR).** To assess whether GRPO training preserves input-output language consistency, we computed the word ratio for both the target language and English. We first remove all LaTeX expressions (e.g., `$...$, \begin{...}, \end{...}`) from the model's output. The remaining text is tokenized using simple regular-expression rules, using Multi-bleu [3], so that punctuation, brackets, and quotes are properly separated. Tokens that consist purely of math expressions, special symbols, or backslash commands are discarded. For each remaining token, we examine its characters to determine whether they belong exclusively to one of several script ranges, such as Hangul (U+AC00–U+D7A3), Latin alphabets (A–Z, a–z), CJK characters (U+4E00–U+9FFF, etc.), or Cyrillic (U+0400–U+04FF). We calculate the *Target word ratio* of a given language by dividing its token count by the total token count. Any token that mixes English letters with another script is labeled as a code-switching token, whose ratio is similarly tracked. This uniform preprocessing and detection pipeline thus enables a quantitative assessment of how models maintain linguistic fidelity in multilingual output. Additionally, we also denote English word ratio as EN WR.

---

[1]Our classification is based on the models' technical reports and cards in Huggingface. The OLMo 2 report only focuses on its English performance, having been trained predominantly on English data. Conversely, the reports for Qwen-2.5 and Llama-3.2 explicitly detail their multilingual capabilities.

[2]https://github.com/huggingface/Math-Verify

[3]https://github.com/moses-smt/mosesdecoder/blob/master/scripts/generic/multi-bleu.perl

Table 1: Accuracy and target-language word ratio for models fine-tuned with GRPO on translated GSM8K. We evaluate on the translated GSM8K and MATH500 test sets. Language codes: **EN** = English, **ZH** = Chinese, **KO** = Korean, **UK** = Ukrainian, **TH**: Thai, **JA**: Japanese. Base Models: **OLMo 2** = OLMo-2-0425-1B-Instruct, **Llama** = Llama-3.2-3B Instruct, **Qwen** = Qwen-2.5-1.5B Instruct. Numbers in parentheses indicate the change relative to these base models. Accuracy (Acc) and target-language word ratio (WR) are reported for each language-model pair.

| Language | Model | GSM8K | | | MATH500 | | |
|---|---|---|---|---|---|---|---|
| | | Target Acc (%) | Target WR (%) | EN WR (%) | Target Acc (%) | Target WR (%) | EN WR (%) |
| ZH | OLMo2 | 59.8 (+34.3) | 0.3 (-75.5) | 80.8 (+73.7) | 17.6(+1.9) | 26.3 (-10.0) | 71.0 (+8.4) |
| | Llama | 69.4 (+7.4) | 94.1 (-1.4) | 8.3 (-0.5) | 38.8 (+1.2) | 77.5 (-0.4) | 18.8 (+0.1) |
| | Qwen | 63.4 (+1.3) | 92.9 (+0.6) | 7.0 (-0.9) | 41.9 (+4.7) | 79.8 (+0.4) | 19.5 (-0.7) |
| KO | OLMo2 | 46.5 (+39.9) | 14.3 (-79.4) | 83.5 (+78.3) | 12.2 (+5.2) | 0.1 (-45.1) | 73.0 (+51.3) |
| | Llama | 61.3 (+14.5) | 82.4 (-8.1) | 14.7 (+7.1) | 28.5 (+7.2) | 70.9 (-17.8) | 21.8 (+16.0) |
| | Qwen | 42.2 (+3.5) | 94.3 (-2.4) | 3.1 (+1.9) | 27.0 (+6.8) | 88.0 (-8.0) | 10.1 (+7.7) |
| UK | OLMo2 | 45.2 (+37.8) | 0.3 (-75.5) | 85.3 (+79.3) | 13.0 (+5.6) | 0.1 (-52.3) | 72.7 (+56.1) |
| | Llama | 70.9 (+17.1) | **0.3** (-82.6) | 96.8 (+80.8) | 47.6 (+12.0) | **5.6** (-72.7) | 93.4 (+73.1) |
| | Qwen | 39.7 (+4.9) | 99.3 (+0.5) | 0.5 (-0.2) | 23.4 (+4.0) | 82.8 (-9.8) | 9.9 (+8.5) |
| TH | OLMo2 | 29.7 (+27.2) | 1.4(-80.3) | 90.5(+81.0) | 9.8(+5.2) | 12.6(-78.4) | 84.7(+76.7) |
| | Llama | 74.6(+16.3) | 84.1(-10.1) | 16.6 (+10.5) | 50.9(+5.8) | 75.4 (-8.1) | 16.8(+10.0) |
| | Qwen | 68.3 (+31.7) | 8.3 (-78.9) | 89.9 (+80.5) | 43.8(+23.9) | 13.1(-76.3) | 83.0(+75.5) |
| JA | OLMo2 | 52.6 (+45.4) | 2.3(-83.9) | 97.0 (+80.1) | 17.0(+2.2) | 7.4(-75.5) | 90.3(+76.9) |
| | Llama | 62.6(+6.6) | 96.4 (-0.6) | 3.3(+0.5) | 37.1(+5.2) | 92.1 (+4.1) | 5.5(+1.8) |
| | Qwen | 43.1 (+7.5) | 95.8(-0.7) | 2.5 (+1.0) | 38.4(+5.7) | 98.9(-0.4) | 1.0(+0.2) |

## 3.2 EXPERIMENTAL VERIFICATIONS OF CROSS-LINGUAL COLLAPSE

For the main study, to verify Cross-lingual Collapse and analyze its behavior, we examine how GRPO-trained models behave on mathematical benchmarks in terms of both *accuracy* and *language fidelity* across five non-English target languages.

**Performance–fidelity trade-off.** Table 1 demonstrates the fine-tuning results across languages (Chinese, Korean, Japanese, Thai, and Ukrainian) and backbones, where selected languages represents distinct alphabet scripts and different pre-training resource levels. RLVR consistently improves target-language accuracy relative to the before fine-tuning (i.e.,SFT models). We can also observe gains in English accuracy, which remains noticeably higher than that of the target languages (Table 6). However, these improvements in the target language often come at the cost of target-language fidelity. The English-dominant LLM, OLMo2-1B, shows the sharpest trade-off across languages. RLVR training yields large accuracy gains but drives the Target word ratio (Target WR) almost to zero and the English Word Ratio (EN WR) toward 80–97%.

On the other hands, multilingual backbones (i.e., Llama-3.2-3B, Qwen-2.5-1.5B) show a resource-sensitive pattern: high-resource Chinese and Japanese preserve language fidelity (Target WR $\geq$ 92%, EN WR $\leq$ 8%), mid-resource Korean shows a moderate drop in Target WR. Low-resource Thai and Ukrainian exhibit distinctive behavior, with significant language drift occurring inconsistently across backbones). Even where Target WR remains high, EN WR is often non-trivial ($\sim$15–20%), which our qualitative analysis traces to short English scaffolding and Latin-script technical tokens embedded in target-language CoTs (Appendix D). This performance-fidelity trade-off persists at larger scales, as detailed in Appendix E, implying that the phenomenon is robust across scales.

Beyond word ratios, we also analyze cross-lingual consistency for a collapsed Chinese GRPO model. As shown in Appendix F, GRPO increases the proportion of problems that the model solves correctly in both English and non-English while reducing cases that are solvable only in English. Combined with the language-fidelity trends in Table 1, this supports the interpretation that the accuracy gains in collapsed models are driven by routing target-language prompts through a stronger English reasoning mode, rather than by genuinely improved target-language reasoning.

These observations reveal a clear trade-off between accuracy and language fidelity under RLVR: accuracy rises while Target WR falls and EN WR rises. We refer to this joint pattern as **Cross-lingual Collapse**—the chain of thought reverts to the pre-training dominant language.

Table 2: Harder training triggers Cross-lingual Collapse in Korean. Qwen-2.5-1.5B Instruct trained on Korean GSM8K alone (*Base*, 1K/2K) preserves target-language fidelity, whereas mixing SimpleRL-Zoo (*Base+Hard*, 2K) collapses Korean word ratio (Target WR) to 14.5%(GSM8K) and 2.1% (MATH500), with accuracy rising to $47.5\%$ and $46.7\%$. On GSM8K, English word ratio (EN WR) also increases, indicating drift toward English.

| Dataset | Steps | GSM8K (KO) | | | MATH500 (KO) | | |
|---|---|---|---|---|---|---|---|
| | | Accuracy (%) | Target WR (%) | EN WR (%) | Accuracy (%) | Target WR (%) | EN WR (%) |
| Base | 1K | 42.3 | 94.3 | 3.1 | 25.7 | 88.0 | 10.1 |
| | 2K | 43.1 | 94.0 | 3.6 | 27.1 | 86.5 | 10.9 |
| Base + Hard | 2K | 47.5 | **14.5** | 80.1 | 46.7 | **2.1** | 87.4 |

(a) GSM8K Accuracy (UK)          (b) Word Ratio (UK)          (c) Word Ratio (UK) w/ Lang loss

Figure 2: Figures 2a–2c compare Llama-3.2-3B Instruct trained with GRPO on the *Ukrainian*-translated GSM8K with and without the language-consistency reward (Lang loss). The language-consistency reward reliably preserves the target-language word ratio, yet it also *dampens* the accuracy gains that GRPO would otherwise deliver. In particular, Figures 2a–2c show that the reward almost completely prevents cross-lingual collapse in the Ukrainian run—though at the cost of a modest drop in performance

## 3.3 TRIGGERING CROSS-LINGUAL COLLAPSE

Building on the trade-off established above, we now unpack *how* the collapse is mechanistically induced, *when* it emerges during training, and *where* it shows up beyond mathematics.

**Difficulty triggers collapse beyond GSM8K.** We verify our hypothesis that problem difficulty accelerate Cross-lingual Collapse even in mid-resource languages (e.g.,KO). We adopt SimpleRL-Zoo as a challenging complement to GSM8K. This increased difficulty widens the reasoning between English and the target language, causing the policy to quickly converge to the more effective English reasoning path. Concretely, for Qwen-2.5-1.5B trained on the *Korean* translation, keeping GSM8K only preserves target-language fidelity after 2K updates (Target WR: GSM8K $94.0\%$, MATH500 $86.5\%$; Table 2). Introducing the harder SimpleRL-Zoo subset collapses the chain-of-thought into English by 2K steps: Target WR falls to 14.5% on GSM8K ($-79.\%5$) and to 2.1% on MATH500 ($-84.4\%$), while accuracy rises to $47.5\%$ on GSM8K and $46.7\%$ on MATH500. Furthermore, we compare SFT approaches in Appendix I. However, SFT approach still show less performance than GRPO approach and the same performance-fidelity trade off.

**Cross-lingual Collapse is initiated during exploration at rollout generation.** Advantage-weighted credit under a correctness-only reward systematically favors English reasoning trajectories, creating a self-reinforcing drift. Figure 3 illustrates for Qwen-2.5-1.5B on Korean GSM8K: exploration often uncovers English CoT continuations that solve the problem more reliably than staying in the target language. Each time such an off-target (English) trajectory succeeds, its advantage is positive, increasing the log-probability of its tokens and shifting future rollouts toward English-Target WR declines while English WR increases. The resulting regime shift—English traces dominating despite non-English prompts—constitutes the rollout-level mechanism behind Cross-lingual Collapse and foreshadows the accuracy jump and fidelity drop observed under harder curricula and high-entropy decoding.

Table 3: Global MMLU-Lite (KO) accuracy and Korean word ratio (Target WR) of CoT outputs for Qwen-2.5-1.5B Instruct trained on GSM8K (KO) under three settings: **Base** (GSM8K only), **Base (w/ Lang loss)** (GSM8K + language-consistency reward), and **Base + Hard** (GSM8K + SimpleRL-Zoo hard curriculum). The hard-curriculum variant achieves the highest accuracy but shows the language drift (lowest Target WR).

|  | **Base** | **Base (w/ Lang loss)** | **Base + Hard** |
|---|---|---|---|
| **Global MMLU** | 31.5 | 31.0 | 33.4 |
| **Target Word ratio** | 71.6 | 75.2 | 23.4 |
| **English Word ratio** | 27.7 | 20.3 | 68.3 |

**Beyond math: domain-general drift.** The other question is whether Cross-lingual Collapse is confined to the mathematical reasoning domain or is a general phenomenon. To investigate this, we evaluated trained models on the Korean question and answer pairs of Global MMLU-Lite (Singh et al., 2024). Specifically, we evaluate three fine-tuning variants of the Qwen2.5-1.5B Instruct model: (1) training with GSM8K (KO), (2) training with GSM8K with a language-consistency loss (Lang loss), and (3) a cross-lingual-collapse setting training with GSM8K and a hard-curriculum dataset (GSM8K + SimpleRL).

As shown in Table 3, the results show a pattern similar to our primary findings on mathematical benchmarks. The cross-lingual-collapse model, fine-tuned with the harder curriculum (GSM8K + SimpleRL), not only achieves the highest performance on MMLU-Lite but also suffers the most severe language drift, with the Korean token ratio in its outputs falling to 23.4%. Conversely, adding the language-consistency reward (Lang loss) preserves a higher Korean token ratio (75.2%) at the cost of a minor dip in performance (31.0). This demonstrates that the trade-off between task accuracy and linguistic fidelity is not confined to mathematics; rather, the pressure to revert to English reasoning for performance gains appears to be a domain-general effect that also holds for general-knowledge tasks.

## 3.4 MITIGATING CROSS-LINGUAL COLLAPSE

Our analyses in §3.3 indicate that cross-lingual collapse is driven by a language-agnostic (accuracy-only) verification reward and exploratory rollouts that discover and reinforce dominant-language reasoning. This observation suggests three complementary mitigation ideas that act at different: (1) **reward shaping** to inject language fidelity into the objective itself; (2) **rollout sampling controls** that constrain exploration so English-only trajectories are less accessible during rollouts; and (3) **training with mixture of multiple languages** that regularize the model's internal arbitration across languages by aligning training with a more balanced linguistic prior.

**Language consistency reward.** Following DeepSeek-AI et al. (2025), we augment the verification reward with an auxiliary signal that favors target-language CoT tokens, as detailed in Appendix H. As shown in Figure 2, we add additional reward in which Llama-3.2-3B is training with GRPO on the Ukrainian GSM8K, once with the language-consistency reward and once without it. In the vanilla setting (Figures 2a–2c, solid line) the model undergoes a full cross-lingual collapse: the share of Ukrainian tokens in its chain of thought drops to almost zero while accuracy rises sharply. Adding the language-consistency reward (dashed line) prevents that collapse—the Ukrainian word ratio stays high—yet the accuracy gain is noticeably smaller. This shows that forcing GRPO to keep the reasoning trace in the target language safeguards linguistic fidelity at the cost of some performance. Furthermore, we compare SFT approaches in Appendix I. However, SFT baselines yield lower performance than GRPO, reinforcing the inherent trade-off between accuracy and fidelity.

These results suggest that during GRPO the model actively probes alternative reasoning paths and, when allowed, gravitates toward high-resource English to maximize reward. Constraining the trace to a non-English language blocks that shortcut, preserving the intended language but sacrificing part of the accuracy gain.

**Adjusting rollout sampling parameters.** Our experiments reveal a consistent dominant-language reversion in chain-of-thought: even under target-language prompts, the word ratio briefly rises and

Table 4: Impact of rollout entropy on Llama3.2-3B with GSM8k(UK) through adjusting top p (Top P) and temperature (Temp.) parameters. The default high-entropy setting (`top_p=1.0`, `Temp=1.0`) maximizes accuracy by allowing the model to revert to high-yield English reasoning paths. Restricting the decoding space with lower `top_p` or `temperature` effectively prevents this language drift, but at the cost of a 5–12 percentage-point drop in accuracy.

| Top P | Temp. | GSM8K (UK) | | | MATH500 (UK) | | |
|---|---|---|---|---|---|---|---|
| | | Accuracy (%) | Target WR (%) | EN WR(%) | Accuracy (%) | Target WR (%) | EN WR(%) |
| 1.0 | 1.0 | 70.9 | 0.3 | 96.8 | 47.6 | 5.6 | 93.4 |
| 0.8 | 1.0 | 64.2 | 81.9 | 11.2 | 35.8 | 83.2 | 15.5 |
| 0.6 | 1.0 | 63.5 | 80.6 | 15.0 | 36.1 | 82.5 | 14.5 |
| 1.0 | 0.8 | 65.6 | 81.2 | 16.0 | 37.4 | 81.0 | 16.9 |

then abruptly flips to the pre-training dominant language (English), coinciding with a sharp accuracy jump—what we term Cross-lingual Collapse. This pattern suggests that reward optimization exploits English as a higher-yield reasoning path in English-centric LLMs. In light of evidence that general language confusion peaks at high-entropy (Marchisio et al., 2024), large-nucleus decoding points and is partially mitigated by lowering temperature and nucleus size, we posit that collapse is a sampling-gated manifestation of the same bias: structural but partially controllable at inference.

As shown in Table 4, reducing temperature or top-p attenuates reversion for Llama-3.2-3B on Ukrainian, though stabilized runs still trail the adding a language consistency reward.

Table 5: Effect of multilingual GRPO training with mix of languages. We train Llama-3.2-3B Instruct on GSM8K with three mixes—UK only, UK+KO, and UK+KO+ZH+EN—and evaluate on Ukrainian GSM8K and MATH500, reporting accuracy and the Target word ratio. Adding Korean alone leaves the model collapsed (near-zero Target WR), whereas a four-language mix largely restores Ukrainian CoT but lowers accuracy.

| Languages | GSM8K(UK) | | | MATH500(UK) | | |
|---|---|---|---|---|---|---|
| | Accuracy (%) | Target WR (%) | EN WR(%) | Accuracy (%) | Target WR (%) | EN WR(%) |
| UK | 70.9 | 0.3 | 96.8 | 47.6 | 5.6 | 93.4 |
| UK, KO | 72.1 | 0.0 | 98.7 | 42.0 | 6.9 | 91.7 |
| UK, KO, ZH, EN | 63.5 | 79.6 | 19.0 | 33.2 | 77.5 | 17.1 |

**Training with multiple languages.** Prior work shows that adding a small set of languages during instruction tuning is more effective than monolingual insturuction tuning (Kew et al., 2024; Chen et al., 2024b; Shaham et al., 2024). We test whether the same idea mitigates Cross-lingual Collapse under RLVR framework. Concretely, we train Llama-3.2-3B Instruct with GRPO on three GSM8K training mixes: (1) Ukrainian only (UK), (2) bilingual (UK+KO), and (3) four-language (UK+KO+ZH+EN). We then evaluate on Ukrainian GSM8K and Ukrainian MATH500, reporting accuracy and the target word ratio of Ukrainian.

As shown in Table 5, adding a single additional language (UK+KO) leaves the model in a collapsed regime on GSM8K. In contrast, training on four languages largely *restores* input–output language consistency on Ukrainian (Target WR $\approx 80\%$ on both test sets), but it *reduces* accuracy relative to the collapsed Ukrainian only (GSM8K: $-7.4$ pp; MATH500: $-14.4$ pp). Thus, multilingual training acts as a crude regularizer against collapse, but introduces a pronounced performance–fidelity trade-off, making it a suboptimal mitigation compared to targeted interventions such as a language-consistency reward and rollout sampling controls.

## 4 DISCUSSION

### 4.1 CROSS-LINGUAL COLLAPSE

The evidence assembled so far paints a coherent picture: (1) Universal Drift. GRPO pushes all models toward the dominant pre-training language, but the speed and severity of that drift scale with

resource level based on the prior work (Wenzek et al., 2020): minimal in high-resource (e.g.,ZH, JA), moderate in mid-resource (e.g., KO), catastrophic in low-resource (e.g.,TH,UK) (Table 1). (2) Difficulty as a Trigger. A mid-resource model that is stable on GSM8K alone collapses after we inject a harder curriculum (Table 2), showing that *task difficulty*, tilts the optimizer toward English reasoning. (3) Reward Design Matters, but Costs Accuracy. Mitigate algorithms partially prevent collapse (Figure 2) yet remove much of GRPO's accuracy gain, implying that the model *strategically* chooses English traces to maximize reward under pressure.

These findings confirm our central claim: GRPO amplifies the linguistic prior that best optimizes reward, and the gap between high- and lower-resource languages widens as tasks grow harder.

### 4.2 Future research direction

Building on the identification and analysis in Sec. 3.2and Sec. 3.3, we designed and evaluated several mitigation algorithms; nevertheless, important limitations persist. Taken together, the experimental results in Sec. 3.4 motivate three research questions to guide future work.

**Persistent accuracy–fidelity trade-off.** Lowering rollout entropy (e.g., via temperature or top-$p$) curbs cross-lingual collapse but also suppresses exploration and hurts accuracy, while higher-entropy sampling does the opposite. This aligns with evidence that broad, diversified search improves reasoning when paired with multi-sample selection or structured exploration—e.g., self-consistency voting and tree-structured search (Wang et al.; Yao et al., 2023)—and with maximum-entropy principles in reinforcement learning that stabilize learning via entropy regularization (Haarnoja et al., 2018; Cui et al., 2025). At the same time, high entropy increases language confusion in multilingual models (Marchisio et al., 2024). A promising direction is therefore to redesign exploration mechanism to keep exploration broad in the semantic space while constraining surface form to the target language.

**Drift is merely incidental or actually the optimizer's "best path" under current objectives.** Our findings are consistent with a reward-shortcut hypothesis under RLVR: high-yield English trajectories discovered during exploration receive positive advantage and become reinforced (Shao et al., 2024; DeepSeek-AI et al., 2025). Rather than fixing a global weight on language fidelity, we propose casting training as constrained or multi-objective RL that explicitly traces the Pareto frontier between accuracy and target-language consistency. Adaptive Lagrangian or primal–dual methods can strengthen the constraint when early warning signals (e.g., a drop in target-language ratio) are detected and relax it otherwise, aiming to block the English shortcut without needlessly sacrificing performance.

**Reconsidering the purpose of interpretable CoT in multilingual settings.** When, if ever, is it acceptable to sacrifice on-language reasoning traces to gain accuracy, and what do we lose in interpretability, auditability, education, and localization when we do? One promising compromise is latent reasoning with target-language summaries: the model reasons internally but must emit concise, on-language plans or explanations for human inspection. Establishing evaluation protocols that jointly reward task accuracy and on-language interpretability will clarify when fidelity should dominate and when performance gains justify off-language traces.

## 5 Related Works

### 5.1 Long Chain-of-Thought Generation

DeepSeek-AI et al. (2025) push the envelope on reinforcement-learning–based reasoning by introducing DeepSeek-R1-Zero, the open-source model trained with pure RL, specifically Group-Relative Policy Optimization (GRPO), without any supervised warm-up, and its follow-up DeepSeek-R1, which adds a small cold-start SFT stage and multi-stage RL to further boost performance. Their study demonstrates that large-scale GRPO can elicit impressive gains on mathematics and coding benchmarks, and that the resulting reasoning patterns can be distilled into much smaller dense models. Notably, the authors briefly report undesirable "language mixing" and readability issues that emerge during RL, suggesting that reward-driven optimization may inadvertently disrupt

linguistic fidelity. However, DeepSeek-R1 focuses almost exclusively on English prompts and does not quantify the extent, or direction, of its language drift. Our work complements these findings by conducting a systematic, multilingual analysis of GRPO and revealing a pronounced *Cross-lingual Collapse*: as RL progresses, chain-of-thought reasoning reverts to the pre-training-dominant language, catastrophically eroding performance in low-resource languages.

## 5.2 MULTILINGUAL INSTRUCTION TUNING

Recent work shows that even a pinch of multilingual data during instruction tuning can unlock substantial cross-lingual generalization in otherwise English-centric LLMs. Shaham et al. (2024) demonstrate that fine-tuning with as few as two to three languages is "necessary and sufficient" to elicit target-language responses across five downstream tasks, with the marginal benefit largely determined by how well that language was covered in pre-training. Complementing this, Kew et al. (2024) find that injecting only 40 non-English instruction–response pairs, or diversifying the tuning mix to merely 2–4 languages, yields instruction-following quality on a par with (or exceeding) monolingual baselines while slashing per-language data by an order of magnitude. Yoo et al. (2024a) demonstrate that incorporating a sufficient amount of code-switched data (combining English and the target language) can effectively adapt an English-centric model, allowing the model to transfer its English-based knowledge into the target. Those studies therefore argue that massive multilingual corpora are not a prerequisite for broad cross-lingual utility; rather, strategically chosen seed languages can act as effective "anchors" that bootstrap transfer to unseen languages. Crucially, neither paper probes how reinforcement-learning–based reasoning objectives interact with this minimalist recipe, leaving open the question of whether such scarce multilingual supervision can withstand the linguistic pressures we observe under GRPO.

## 5.3 MULTILINGUAL REASONING

Mechanistic analyses show that multilingual LLMs are not language-neutral: logit-lens (Schut et al., 2025) studies find models like Llama-3.1 route concepts through an English-centered space even for non-English prompts, and steering vectors learned in English transfer more robustly; circuit tracing of Claude 3.5 Haiku reveals language-agnostic subcircuits cooperating with language-specific pathways, yet English often dominates when languages compete (Lindsey et al., 2025). Building on this asymmetry, two families of methods explicitly leverage English reasoning to boost multilingual performance: (i) pivot-translation approaches translate questions or intermediate steps into English to exploit stronger reasoning priors and tools, then map solutions back to the target language (Zhu et al., 2024; Chen et al., 2024a; Yoon et al., 2024); and (ii) cross-lingual preference alignment aligns step-level choices across languages via preference optimization (She et al., 2024). These works chiefly optimize outcomes rather than explain failure modes. In contrast, we identify when and why Cross-lingual Collapse emerges in RL-based reasoning and link it to English-biased latent computation, offering a diagnostic lens complementary to cross-lingual consistency work and clarifying how language-specific reasoning abilities emerge—and sometimes fail—under optimization pressure.

## 6 CONCLUSION

This study uncovers and characterizes **Cross-lingual Collapse**: when trained with reinforcement learning with verifiable reward and long chain-of-thought, large language models (LLMs) increasingly route their reasoning through the pre-training–dominant language as accuracy rises. Across five target languages and multiple backbones, we observe a clear resource-sensitivity gradient: negligible drift in high-resource Chinese/Japanese, moderate in mid-resource Korean, and severe collapse in low-resource Thai/Ukrainian, with English-centric backbones collapsing fastest. The effect persists beyond mathematics. A language-consistency reward, entropy reduction at rollout time (e.g., lower temperature), and multilingual RLVR all preserve target-language traces to varying degrees, but each incurs a measurable accuracy cost; even broad multilingual mixes largely restore on-language CoT while lowering scores. These results reveal a persistent *performance–fidelity* trade-off. We view this phenomenon as a natural consequence of English-dominant pre-training and argue that securing linguistic diversity during pre-training is a necessary (though not always sufficient) condition for maintaining language fidelity in long CoT settings.

# 7 REPRODUCE STATEMENT

In order to ensure the reproduceability of the project, we describe details hyperparameter configurations and dataset creation pipeline described in Sec. 3.1. We will release the datasets and code ,including configuration files and reproduction scripts, in a public GitHub repository upon publication to enable end-to-end replication of our results.

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

# A    MAIN TABLE WITH ENGLISH-LANGUAGE ACCURACY

Table 6: Accuracy and target-language word ratio for models fine-tuned with GRPO on translated GSM8K. We evaluate on the translated GSM8K and MATH500 test sets. Language codes: **EN** = English, **ZH** = Chinese, **KO** = Korean, **UK** = Ukrainian. Model keys: **OLMo 2** = OLMo-2-0425-1B-Instruct, **Llama** = Llama-3.2-3B Instruct, **Qwen** = Qwen-2.5-1.5B Instruct. Numbers in parentheses indicate the change relative to the corresponding non-fine-tuned baseline. Accuracy (Acc) and target-language word ratio (WR) with languages and models arranged as rows.

| Language | Model | GSM8K | | | MATH500 | | |
|---|---|---|---|---|---|---|---|
| | | Target Acc (%) | Target WR (%) | EN Acc (%) | Target Acc (%) | Target WR (%) | EN Acc (%) |
| ZH | OLMo2 | 59.8 (+34.3) | 0.3 (-75.5) | 74.8 (+4.0) | 17.6(+1.9) | 26.3 (-10.0) | 21.4 (+0.7) |
| | Llama | 69.4 (+7.4) | 94.1 (-1.4) | 83.5 (+3.4) | 38.8 (+1.2) | 77.5 (-0.4) | 50.3 (+1.8) |
| | Qwen | 63.4 (+1.3) | 92.9 (+0.6) | 77.9 (+4.0) | 41.9 (+4.7) | 79.8 (+0.4) | 55.7 (+7.5) |
| KO | OLMo2 | 46.5 (+39.9) | 14.3 (-79.4) | 73.1 (+2.3) | 12.2 (+5.2) | 0.1 (-45.1) | 22.2 (+1.5) |
| | Llama | 61.3 (+14.5) | 82.4 (-8.1) | 81.6 (+1.5) | 28.5 (+7.2) | 70.9 (-17.8) | 49.6 (+1.1) |
| | Qwen | 42.2 (+3.5) | 94.3 (-2.4) | 74.1 (+0.2) | 27.0 (+6.8) | 80.3 (-12.3) | 54.1 (+5.9) |
| UK | OLMo2 | 45.2 (+37.8) | 0.3 (-75.5) | 73.7 (+2.9) | 13.0 (+5.6) | 29.8 (-57.4) | 21.6 (+0.7) |
| | Llama | 70.9 (+17.1) | **0.3** (-97.6) | 80.8 (+0.6) | 47.6 (+12.0) | **5.6** (-72.7) | 51.2 (+1.7) |
| | Qwen | 39.7 (+4.9) | 99.3 (+0.5) | 75.4 (+1.6) | 23.4 (+4.0) | 82.8 (-9.8) | 51.2 (+3.0) |

# B    TRANSLATED DATASET DETAILS

To ensure high translation quality, we re-translated the English source with GPT-4o, a model that exhibits near-professional performance across many language pairs (Yan et al., 2024; Blain et al., 2023). After each pass we filtered candidates with xCOMET (Guerreiro et al., 2024): only hypotheses that exceeded the Flores dev-set mean for the target language were retained; sentences falling below the threshold were re-translated.

Moreover, to validate training data quality, we conducted an additional sanity check. We normalized all numerals across languages to Arabic digits and verified via regex that every number in the English source appeared unchanged in the translation. This process ensures that all translations properly preserve the original Arabic digits and equations.

## C ROLLOUT GENERATION EXAMPLE

**Question (Korean)**

비고의 나이는 그의 남동생이 2살일 때 남동생 나이의 두 배보다 10살 더 많았습니다. 만약 그의 남동생이 현재 10살이라면,

(Vigo was 10 years older than twice his younger brother's age when the brother was 2 years old. If the younger brother is now 10)

그들 의 나이 합은 얼마입니까? 차근차근 단계별로 논리적으로 풀어주세요, 최종 답은 \boxed{} 안에 넣어주세요.

(what is the sum of their current ages? Please solve the problem step by step, showing your reasoning logically, and place the final answer inside \boxed{}.)

**Rollout Examples during Training**

1. 남동생의 현재 나이를 확인합니다. 문제에서 남동생이 현재 10살이라고 주어졌습니다.
(1. Confirm the younger brother's current age. According to the problem, he is now 10 years old.)
[...]
남동생이 지금 10살이므로, 비고는 10 + 12 = 22살입니다. 4. 두 사람의 현재 나이 합을 구합니다. 10 + 22 = 32
따라서, 두 사람의 나이 합은 \boxed{32}
(Since the younger brother is 10, Vigo is 10 + 12 = 22 years old. 4. Calculate the sum of their current ages:
10 + 22 = 32. Therefore, the sum of their ages is \boxed{32})

→ Target-language Reward

Let's break this down step by step:\n\n1. **Identify the current age of the younger brother:**\n - The problem states that the younger brother is currently 10 years old.
[...]
The younger brother is 10 years old.\n - The older brother is 22 years old.\n - The sum of their ages is \\(10 + 22 = 32\\).\n\nTherefore, the sum of their ages is \boxed{32}

→ Non-target-language Reward

Figure 3: Rollout examples from GRPO training of Qwen-2.5 1.5B on the Korean-translated GSM8K. Observe that the model often arrives at the right answer via English reasoning (*non-target language*); because any correct answer earns full reward, repeated reinforcement of such off-language traces gradually shifts the chain-of-thought word ratio away from Korean.

## D QUALITATIVE ANALYSIS OF LANGUAGE FIDELITY IN REASONING TRACES

We analyze the inference results of fine-tuned models to study patterns in reasoning traces regarding language fidelity aspect. We categorize these distinct patterns: **(1) Cross-lingual Collapse**, where the model generates both the reasoning trace and the final response in the dominant language (i.e., English) regardless of the input language; **(2) English Scaffolding and Terminology**, where the model generates the response in the target language but keeps English for structural markers (e.g., "Step 1") or technical terminologies; and **(3) Latin Alphabet Tokens**, where Latin-script represents algorithm names, function identifiers, or unit abbreviations.

Here, patterns (2) and (3) are the primary reasons why the English Word Ratio (EN WR) remains non-zero. Unlike cross-lingual collapse, these patterns result in a mixture of the target language and English words. In contrast, models that maintain high multilingual fidelity (e.g., Qwen on Ukrainian tasks) successfully localize both scaffolding and technical terms, thereby achieving a near-zero EN WR.

Inference Example 1: Cross-lingual Collapse

**Question** : 花瓶にバラが4本あります. ダリアはバラより7本多いです.花瓶の中に花は全部で何本ありますか?

**Response** :

To determine the total number of flowers in the vase, we need to find the number of each type of flower and then add them together. Let's break it down step by step.
1. **Number of roses in the vase:** - According to the problem, there are 4 roses in the vase.
2. **Number of daisies in the vase:** - The problem states that the number of daisies is 7 more than the number of roses. Since there are 4 roses, we need to add 7 to find the number of daisies.
$$\text{Number of daisies} = 4 + 7 = 11$$
3. **Total number of flowers in the vase:**
- Now, we need to find the total number of flowers by adding the number of roses and the number of daisies.
$$\text{Total number of flowers} = \text{Number of roses} + \text{Number of daisies} = 4 + 11 = 15$$

Therefore, the total number of flowers in the vase is $\boxed{15}$ .

---

Inference Example 2: Scaffolding or terminology inside CoT

**Question** : 카를로스는 레몬 나무를 심고 있습니다. 나무를 심는 데에는 90달러가 듭니다. 이 나무에서는 매년 7개의 레몬이 열리고, 레몬은 개당 1.5달러에 팔 수 있습니다. 나무에 물과 비료를 주는 데에는 매년 3달러가 듭니다. 카를로스가 이 나무에서 순수익을 내기 시작하려면 몇 년이 걸릴까요?

**Response** :
1. **Initial Investment**: 카를로스는 레몬 나무를 심는데 90달러가 들었습니다.
2. **Annual Revenue**: 매년 7개의 레몬이 열리고, 각각 1.5달러에 팔 수 있습니다. 따라서 매년 수익은 $7 \times 1.5 = 10.5$달러입니다.
3. **Annual Expenses**: 매년 3달러의 비용이 듭니다.
4. **Net Annual Profit**: 매년 수익에서 비용을 뺀 결과는 $10.5 - 3 = 7.5$달러입니다.
5. **Time to Break Even**: 초기 비용을 빼고, 매년 수익을 빼면, 카를로스가 수익을 내기 시작하려면 $90/7.5 = 12$년이 걸립니다.

최종 답: $\boxed{12}$ 년

---

### Inference Example 3: Using english Abbreviation term

**Question** : 如果任意一点 $A$ $B$ $C$ 或 $D$ 出, 其他三点各一次, 最短的行程距是多少?

**Response** :

了到任意一点 $A$ $B$ $C$ 或 $D$ 出, 其他三点各一次的最短行程距, 我可以考化一, 其中每点代表一点, 每代表一距 我需要到任意一点出, 其他三点, 最后回到原点的最短路。

根据目出的, 我可以看到:
- $A$ 到 $B$ 的距是 3
- $A$ 到 $C$ 的距是 6
- $A$ 到 $D$ 的距是 5
- $B$ 到 $C$ 的距是 6
- $B$ 到 $D$ 的距是 4
- $C$ 到 $D$ 的距是 5

了到最短路, 我可以使用中的最短路算法, 如Dijkstra算法 但是, 由于是一的, 我可以直接算。

首先, 我 $A$ 出, $B$ $C$ 和 $D$ 各一次, 然后回到 $A$ 最短路可以通以下步算:
1. $A$ 到 $B$ 的距是 3。
2. $B$ 到 $C$ 的距是 6。
3. $C$ 到 $D$ 的距是 5。
4. $D$ 到 $A$ 的距是 5。

因此, 最短路是 $A \rightarrow B \rightarrow C \rightarrow D \rightarrow A$, 距是 $3 + 6 + 5 + 5 = 19$。

所以, 最短的行程距是 $\boxed{19}$。

Table 7: GRPO training results with 7B or 8B backbones on translated GSM8K dataset. We report accuracy, target-language word ratio (Target WR), and English word ratio (EN WR). Numbers in parentheses denote the change relative to the corresponding backbone.

| Model | Language | Accuracy (%) | Target WR (%) | EN WR (%) |
|---|---|---|---|---|
| OLMo-2-7B-SFT | ZH | 83.3 (+33.6) | 0.3 (-80.9) | 87.7 (+83.6) |
| OLMo-2-7B-SFT | KO | 71.1 (+58.3) | 0.2 (-82.1) | 89.1 (+81.8) |
| Llama-3.1-8B-Instruct | KO | 76.2 (+7.9) | 92.4 (-4.7) | 6.8 (+4.3) |
| Llama-3.1-8B-Instruct | ZH | 91.3 (+8.3) | 90.8 (+6.8) | 7.8 (-5.4) |

## E    LARGER MODEL EXPERIMENTS

To verify whether our findings extend to larger models, we conduct additional experiments on two larger backbones: OLMo-2-7B-SFT and Llama-3.1-8B-Instruct. We follow the same experimental settings and evaluation metrics as described in Section 3.1, training on translated GSM8K in Chinese (ZH) and Korean (KO).

Table 7 demonstrate the results. For OLMo-2-7B, GRPO dramatically improves GSM8K accuracy in both Chinese and Korean, but Target WR drifts to almost zero while the English word ratio rises to nearly 90%. This reproduces the collapse behavior we observed for the smaller OLMo backbone in Table 1. On the other hands, Llama-3.1-8B-Instruct have improvment while preserving high target-language fidelity: the target-language word ratio remains above 90% in both Chinese and Korean, and English WR retains below 10%. These results indicate that cross-lingual collapse persists at larger scales and strongly depends on the backbone.

Table 8: Cross-lingual consistency between English and Chinese on OLMo-2-7B with Chinese dataset. Each column corresponds to one of three correctness patterns on the same math problem: (✓, ✓) for both English and Chinese correct, (✓, ✗) for English-only success, and (✗, ✓) for Chinese-only success. Larger proportion of the agreement case ((✓, ✓)) and smaller proportion of the disagreement cases ((✗, ✓) and (✓, ✗)) indicates stronger cross-lingual consistency.

| | Cross-lingual Consistency | | |
|---|---|---|---|
| | (✓, ✓) ↑ | (✓, ✗) ↓ | (✗, ✓) ↓ |
| OLMo-2(7B) + GRPO (ZH) | **79.7** | **9.9** | 3.6 |
| OLMo-2(7B) | 46.4 | 35.8 | **3.1** |

## F  CROSS-LINGUAL CONSISTENCY ANALYSIS IN COLLAPSED MODELS

To quantify how GRPO affects alignment between English and Chinese under the Cross-lingual collapse case, we measure cross-lingual consistency on GSM8K for OLMo-2-7B-SFT before and after GRPO training on GSM8K(ZH). For each test item, we query the model twice: once with the original English question and once with the Chinese translation, using the same evaluation. We then categorize each item according to whether the model is correct (✓) or incorrect (✗) under the English and Chinese prompts, computing the proportion of three cases: (✓,✓), (✓,✗) and (✗,✓).

Table 8 reports the resulting ratios. For the Instruct backbone, only 46.4% of items are solved in both languages, while 35.8% are correct in English but wrong in Chinese. After fine-tuning with GRPO, the collapsed model displays much stronger cross-lingual alignment: the (✓,✓) category increases to 79.7% and the (✓,✗) category drops to 9.9%, while the (✗,✓) category remains small (3-4%) for both models. Thus, GRPO largely eliminates cases where the model can solve a problem only in English but not in the target language.

Consistent with this view, among the problems that the GRPO model solves under Chinese prompts, **95.7%** are also solved correctly when the same model is run on the English input. Combined with the language-fidelity results in Table 1, this supports the interpretation that accuracy gains in the collapsed model come from routing the target language prompts through the model's stronger English reasoning, rather than from improved target-language reasoning capabilities

## G  FURTHER TRAINING OF DISTILLED LRMS

As depicted in Figure 4, we apply a second round of GRPO to the DeepSeek-R1–Distilled Qwen to test whether continued fine-tuning can correct the entrenched reasoning bias. The results reveal a steep decline in the target-language word ratio, indicating that the phenomenon is difficult to reverse.

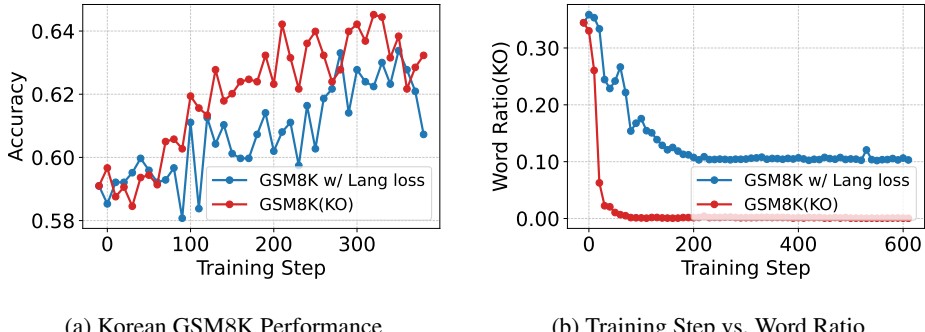

(a) Korean GSM8K Performance          (b) Training Step vs. Word Ratio

Figure 4: We continued GRPO fine-tuning of the DeepSeek-R1-Distill Qwen model on the Korean-translated GSM8K dataset to encourage Korean chain-of-thought reasoning. As Figure 4b shows, the distilled model still exhibits cross-lingual collapse during training.

Table 9: Ablation of language-consistency interventions for Llama-3.2-3B Instruct on Ukrainian GSM8K. All three approaches substantially improve target-language word ratio (Target WR) relative to vanilla GRPO, at the cost of lower accuracy.

| Model / Setting | GSM8K (Acc %) | Target WR (%) | EN WR (%) |
|---|---|---|---|
| GRPO (UK) | 70.9 | 0.3 | 96.8 |
| GRPO + Language proportion reward | 62.5 | 81.6 | 16.4 |
| GRPO + Language threshold reward | 62.1 | 85.0 | 11.9 |
| GRPO + Rollout-level filtering | 63.8 | 88.0 | 13.0 |

## H    ABLATIONS OF LANGUAGE-CONSISTENCY INTERVENTIONS

To analyze how different language-consistency interventions affect cross-lingual collapse at the rollout level, we start from the Ukrainian GSM8K setting in Sec. 3.1. For each sampled rollout $y$ we compute its target-language word ratio $\text{TargetWR}(y)$ using the script-based preprocessing pipeline described in Sec. 3.1.

**Language-consistency reward.**    We study two ways of injecting language fidelity into the scalar reward:

1. **Proportion reward.** We add the target-language ratio directly to the correctness reward:

$$r_{\text{total}}(y) = r_{\text{corr}}(y) + \lambda \cdot \text{TargetWR}(y),$$

   where $\lambda$ represents the weight of the language reward. We set $\lambda$ to 0.5 in all experiments.

2. **Threshold reward.** We add a fixed bonus of $0.5$ if the rollout is mostly in the target language and sufficiently long:

$$r_{\text{total}}(y) = r_{\text{corr}}(y) + \lambda \cdot \mathbf{1}[\text{TargetWR}(y) \geq 0.5 \ \wedge \ \text{len}(y) > 10],$$

   where $\text{len}(y)$ counts non-LaTeX tokens. The length constraint avoids degenerate short but the target language responses. We also set $\lambda$ to 0.5 in the experiment.

We denote the proportion reward and threshold reward as Language proportion reward and Lang threshold reward in Table 9, respectively. Note that the model with the language reward in Fig. 2 utilizes the proportion reward as an auxiliary reward.

**Rollout-level filtering.**    We also evaluate an alternative that acts on rollout sampling level rather than on the reward. For each trajectory $y$ we compute the English word ratio $\text{ENWR}(y)$. If $\text{ENWR}(y) \geq 0.5$, we discard $y$ and resample until the number of valid samples is filled. This procedure filters out trajectories whose reasoning has already collapsed into predominantly English. We denote this approach as rollout-level filtering in Table 9

**Results.**    Table 9 reports results for Llama-3.2-3B Instruct trained with GRPO on Ukrainian GSM8K. All three interventions substantially increase the target-language word ratio (from $0.3\%$ under vanilla GRPO to $82$–$88\%$) and reduce the English word ratio to around $12$–$16\%$. However, they also lower GSM8K accuracy from $70.9\%$ to $62$–$64\%$, i.e., by roughly 7–9 percentage points relative to the correctness-only baseline. Among the three, English-response detection offers the best accuracy–fidelity trade-off, but it does not remove the underlying performance–fidelity tension highlighted in Sec. 3.4.

Table 10: Qwen-2.5-1.5B Instruct on Korean GSM8K and MATH500 under GRPO and supervised fine-tuning (SFT). We report accuracy, target-language word ratio (Target WR), and English word ratio (EN WR) for two curricula: Base (GSM8K(KO) only) and Base+Hard (GSM8K(KO) + SimpleRL-Zoo)

| Dataset | GSM8K (KO) | | | MATH500 (KO) | | |
|---|---|---|---|---|---|---|
| | Accuracy (%) | Target WR (%) | EN WR (%) | Accuracy (%) | Target WR (%) | EN WR (%) |
| GRPO (Base) | 43.1 | 94.0 | 3.6 | 27.1 | 86.5 | 10.9 |
| GRPO (Base + Hard) | 47.5 | **14.5** | 80.1 | 46.7 | **2.1** | 87.4 |
| SFT(Base) | 40.9 | 97.0 | 2.8 | 24.0 | 94.2 | 4.1 |
| SFT(Base + Hard) | 42.5 | 96.8 | 3.1 | 30.9 | 94.5 | 5.4 |

## I  ADDITIONAL SUPERVISED FINE-TUNING BASELINES

To complement our RLVR results, we conduct an additional supervised fine-tuning (SFT) study on Korean. Our goal is to test whether enriched target-language math supervision can narrow the accuracy gap to GRPO-based models and mitigate Cross-lingual Collapse.

**Experimental setup.**  For a fair comparison with the GRPO models, we reuse the same math datasets as in the main experiments: GSM8K and SimpleRL-Zoo. We build a Korean SFT corpus by translating the original English CoTs of the dataset into Korean, following exactly the same translation pipeline in Appendix B.

We consider two SFT configurations for Qwen-2.5-1.5B Instruct: (i) SFT on Korean GSM8K only(i.e.,SFT(Base)), and (ii) SFT on the union of Korean GSM8K and the hard curriculum (SimpleRL-Zoo) with their translated responses (SFT(Base+ Hard). In both configurations, we train for two epochs to mitigate overfitting, using the same optimizer, learning rate, and other hyperparameters as in the GRPO runs.

**Results.**  Table 10 demonstrates that the accuracy of SFT variants increase while keeping Target WR high, but they still underperform the GRPO-based models on GSM8K dataset. Moreover, GRPO (Base + Hard) model attains the highest accuracy at the cost of reduced Target WR and increased EN WR. This result still shows the accuracy–language fidelity trade-off.

