# OpenReview forum: "When Verifiable Rewards Switch the Language: Cross-Lingual Collapse in Chain-of-Thought"
_ICLR.cc/2026/Conference — Submitted to ICLR 2026_

### Official Review · Reviewer_o4RZ · 2025-10-29

**Soundness:** 2
**Presentation:** 2
**Contribution:** 2
**Rating:** 2
**Confidence:** 4

**Summary:**

This paper investigates the widely acknowledged phenomenon of "cross-lingual collapse", demonstrating that multilingual LLMs experience degraded reasoning in non-English languages, primarily correlating this effect with factors like disproportionate token usage. Through controlled experiments and reward-based interventions, it analyzes the causes, effects, and partial mitigations of this language drift during reasoning tasks.

**Strengths:**

The paper introduces an observation about cross-lingual collapse in multilingual math reasoning. It attempts to contribute a mechanistic investigation by seeking to attribute this known phenomenon to specific internal factors like token usage or representational imbalances, even if this analysis is ultimately superficial.

**Weaknesses:**

- The central premise, the "cross-lingual collapse" of reasoning capabilities, is a known problem frequently discussed across numerous influential papers concerning multilingual and English-centric models. Especially it has been clearly pinpointed in the paper of GRPO [1]. The authors waste significant space describing a phenomenon without proposing a truly novel mechanism, a new theoretical framework, or a breakthrough solution.

- The experiments are restricted to relatively small models (up to 3B parameters) that are known to exhibit unstable behavior, which limit the generalizability of the findings to larger models commonly used in practice.

- Only three languages are used (Chinese, Korean, and Ukrainian), which constraints the generality of the conclusions across typologically diverse languages and language‐resource levels. It lacks the inclusion of other script languages such as, Arabic, Latin, Bengali, Tamil and others.

- The study uses translated versions of standard math datasets into target languages. Translation quality and cultural or linguistic adaptation issues may cloud the results. It is unclear whether the collapse is caused by translation artifacts, as the solutions to math problems may depend on specific linguistic or cultural contexts.

- There is no thorough human evaluation of whether the reasoning in the target language is meaningful, or whether the drift to dominant language improves or harms reasoning quality in the target language context.

- The findings might be narrowly applicable to math reasoning tasks, and may not generalize to other reasoning domains such as commonsense reasoning, logical reasoning, or real-world problem solving. Please

- For the language consistency reward, the paper acknowledges the drop in performance and the mitigation of collapse, but gives limited analysis of when the trade-off might be acceptable.

- The manuscript is poorly written, likely suffering from overly complex jargon intended to mask simple observations


[1] Guo, Daya, et al. "Deepseek-r1: Incentivizing reasoning capability in llms via reinforcement learning." arXiv preprint arXiv:2501.12948 (2025).

**Questions:**

- Does the languages similarity impact the conclusions?

- How to ensure the quality of the translated datasets? Is there any human evaluation or the results of automatic metrics?

---

> ### Author Response · Authors · 2025-11-21
>
> Dear reviewer o4RZ, We are grateful for the valuable feedback provided by the reviewer. In this response, we thoroughly address each of your comments and concerns.
>
> ---
>
> **[Weakness 1] The central premise, the "cross-lingual collapse" of reasoning capabilities, is a known problem frequently discussed across numerous influential papers concerning multilingual and English-centric models. Especially it has been clearly pinpointed in the paper of GRPO [1]. The authors waste significant space describing a phenomenon without proposing a truly novel mechanism, a new theoretical framework, or a breakthrough solution.**
>
> We agree with the reviewer that prior technical reports from DeepSeek and others have already noted that RLVR-trained models often produce reasoning traces that are not consistently fluent or “good” English. Our goal is not to claim that such irregularities were previously unknown, but to make this phenomenon precise, measurable, and configurable in a multilingual setting. Concretely, we (i) coin and operationalize Cross-lingual Collapse as a joint pattern of rising task accuracy and systematic drift of CoT into the pre-training–dominant language, quantified via accuracy and target/English word-ratio metrics; (ii) show that this language switch reliably appears in standard, lab-scale RLVR setups; and (iii) analyze both the factors that amplify or attenuate the drift and the rollout-level mechanisms that drive it.
>
> While the DeepSeek [1] reports qualitatively mention language mixing and degraded readability, they do not provide a formal definition of the phenomenon, characterize when and why it appears under non-English prompts, or systematically examine how it can be mitigated. In contrast, we deliberately work in a controlled, reproducible setting to isolate Cross-lingual Collapse and study simple, intuitive interventions (language-consistency reward, decoding controls, multilingual GRPO) that disentangle its causes and reveal its persistent performance–fidelity trade-off. As we note in the paper, this trade-off ultimately stems from English-dominant pre-training priors and thus cannot be fully removed; however, making the failure mode explicit provides actionable insight for practitioners who need multilingual reasoning models with predictable and controllable language behavior.
>
> [1] Guo, Daya, et al. "Deepseek-r1: Incentivizing reasoning capability in llms via reinforcement learning." arXiv preprint arXiv:2501.12948 (2025).
>
> ---
>
> **[Weakness 2] The experiments are restricted to relatively small models (up to 3B parameters) that are known to exhibit unstable behavior, which limit the generalizability of the findings to larger models commonly used in practice.**
>
> We agree that it is valuable to validate that our conclusions are not an limited only small models. To address this, we run additional experiments with up to 3B backbones, OLMo-2-7B-SFT and Llama-3.1-8B-Instruct.  Due to constraints, we focused these large‑model experiments on two representative backbones and two non‑English languages  As shown in the below result table, we observe the same patterns of cross‑lingual collapse and backbone‑dependent behavior as in our 1B–3B models.
>
> Specially, for OLMo2‑7B, GRPO dramatically improves accuracy in both Chinese(Zh) and Korean(Ko), but the target‑language word ratio collapses in near zero, while English WR ratio is almost 97%. This is exactly the similar pattern in Table 1. In LLaMA3-8B, Chinese GSM8K accuracy increases while maintaining high language-fidelity (i.e.,TR WR).
>
> We will add this new experiment to the revised manuscript until discussion period.
>
>
>
> | Model |Language | GSM8K | TR(WR) | EN(WR) |
> |---------|--------|------|-----|---|
> |OLMO2-7B| zh | 83.3 (+33.6) |0.3(-80.9) | 87.7(+83.6) |
> |OLMO2-7B | ko |71.1 (+58.3)|0.2 (-82.1) | 89.1(+81.8)|
> |LLaMA3-8B| ko| 76.2 (+7.9)| 92.4(-4.7)| 6.8(+4.3)|
> |LLaMA3-8B| zh| 91.3 (+8.3)| 90.8 (+6.8) | 7.8( -5.4)|

---

> ### Author Response · Authors · 2025-11-21
>
> **[Weakness 3] Only three languages are used (Chinese, Korean, and Ukrainian), which constraints the generality of the conclusions across typologically diverse languages and language‐resource levels. It lacks the inclusion of other script languages such as, Arabic, Latin, Bengali, Tamil and others.**
>
> We appreciate the comment about typological coverage. To probe generality beyond the three languages in our experiment, we conduct additional experiments on Japanese and Thai languages. For each language we fine‑tuned OLMo‑2 and Llama‑3.2‑3B with RLVR and evaluated accuracy together with the target‑language word ratio. We also observed model‑dependent cross‑lingual collapse: OLMo‑2 showed drift into English, whereas Llama‑3.2‑3B exhibited smaller performance gains and correspondingly weaker drift. These results reproduce the accuracy–fidelity trade‑off reported in our paper and extend it to additional scripts, supporting that the phenomenon is not confined to a particular language.
>
> | Model |Language | GSM8K | TR(WR) | EN(WR) |
> |---------|--------|------|-----|----|
> |OLMO2-1.5B| ja |52.61 (+45.4)  | 2.3(-83.9) | 97.0 (+80.1)|
> |OLMO2-1.5B| th | 29.7(+27.2) | 1.4(-80.3)|  90.5( 81.0) |
> |Qwen-1.5B| ja | 43.1 (+7.5)| 95.8(-0.7)| 2.5 (+1.0)|
> |Qwen-1.5B | th| 68.3 (+31.7)|8.3 (-78.9) |89.9 (+80.5)|
>
>
> ---
>
> **[Weakness 4] The study uses translated versions of standard math datasets into target languages. Translation quality and cultural or linguistic adaptation issues may cloud the results. It is unclear whether the collapse is caused by translation artifacts, as the solutions to math problems may depend on specific linguistic or cultural contexts.**
>
> The cross‑lingual collapse we observe is not an artifact of translation. Mathematics problems are known as as culture‑invariant problems [2], and we ensured translation quality with a two‑stage pipeline (GPT‑4 translation plus xCOMET filtering) detailed in Appendix B. To further validate our dataset, we conducted two additional experiments.
>
> (1) Sanity checks of tanslaion dataset: We normalized all numerals of samples per languages to Arabic digits and verified through regex that every number in the English source appears unchanged post‑translation. As a result, all of our trasnlation keeps arabic digits and its equation.
>
> (2) Using other translated dataset: We reproduced the phenomenon using an independently translated Chinese GSM8K from prior work[3]. OLMo‑2 fine‑tuning exhibits the same cross-linugal collapse. This result implies that our observations are robust and not limited to our dataset.
>
> | Model | GSM8K | TR(WR) | EN(WR) |
> |---------|--------|------|-----|
> | Olmo2-1.5B| 57.2 | 4.8 (-70.9)| 86.8 (+79.7)|
>
> [2] Singh et al. Global MMLU: Understanding and Addressing Cultural and Linguistic Biases in Multilingual Evaluation.
> [3] Chen et al. Breaking Language Barriers in Multilingual Mathematical Reasoning: Insights and Observations
>
> ----
>
> **[Weakness 5] There is no thorough human evaluation of whether the reasoning in the target language is meaningful, or whether the drift to dominant language improves or harms reasoning quality in the target language context.**
>
> To address the concern about the reasoning quality of our fine‑tuned models, we will add examples of  target‑language reasoning traces in the revision by the rebuttal deadline. They illlustrate show whether the models produce coherent step‑by‑step chains in the target language in collapased models or not.
>
> Our paper investigate Cross-lingual Collapse in the context of RLVR. We characterize the phenomenon, pinpoint the underlying amplifiers at the rollout anlysis, and report the limits of mitigation strategies. Our experiments, which track accuracy and WR across various LLMs and languages, reveal that performance gains from RLVR come at a cost: the chain of thought systematically collapses into pretraining dominant language (i.e.,English).
>
> ---

---

> ### Author Response · Authors · 2025-11-21
>
> **[Weakness 6] The findings might be narrowly applicable to math reasoning tasks, and may not generalize to other reasoning domains such as commonsense reasoning, logical reasoning, or real-world problem solving. Please**
>
> Our findings can be observed in the common sense reasoning task. Specifically, we evaluated the same GRPO‑trained models on Global MMLU‑Lite (KO) in Sec 3.3, a general‑knowledge/common sense benchmark. The model that collapses on math under the harder curriculum shows the same English‑drift on GMMLU‑Lite: it achieves the best accuracy (33.4) while its chain‑of‑thought is mostly English (Target WR 23.4%, EN WR 68.3%). This result shows that cross‑lingual collapse extends beyond mathematic problems to commonsense reasoning tasks.
>
> ---
>
> **[Weakness 7] For the language consistency reward, the paper acknowledges the drop in performance and the mitigation of collapse, but gives limited analysis of when the trade-off might be acceptable.**
>
> We appreciate the reviewer’s point. Our intent with the language-consistency reward was primarily diagnostic, not to fully solve collapse or prescribe a single “acceptable” operating point. As discussed in Section 3.4 and Section 4.2, we use this intervention to make the performance–fidelity trade-off concrete: forcing the model to stay in the target language reliably mitigates collapse, but the same mechanism that drives collapse (reward-seeking toward high-resource English) also explains why accuracy drops when that shortcut is blocked. Our contribution is to (i) reveal and quantify this structural trade-off under RLVR and (ii) interpret it through the lens of cross-lingual collapse, rather than to claim that we have fully resolved it.
>
> ---
>
> **[Question 1] Does the languages similarity impact the conclusions?**
>
> We thank the reviewer for this insightful suggestion. We agree that analyzing the correlation between linguistic similarity to English and the severity of Cross-lingual Collapse is a compelling direction for future work.
>
> In particular, we believe examining script (alphabet) similarity—or orthographic overlap—is crucial. We hypothesize that languages sharing the Latin alphabet (or sharing significant sub-word tokens with English) may lower the barrier for language switching, effectively acting as a bridge that accelerates the drift into English compared to languages with disjoint scripts.
>
> ---
>
> **[Question 2] How to ensure the quality of the translated datasets? Is there any human evaluation or the results of automatic metrics?**
>
> We ensured translation quality through a two‑stage pipeline and then ran additional validation addressing Weakness 4.
>
> As detailed in Appendix B, we first generated translations with GPT‑4o, then applied xCOMET[4] quality‑estimation filtering. We retained only segments whose xCOMET  score exceeded the FLORES dev‑set mean for the target language, while re-translating those falling below the threshold. xCOMET is trained to correlate with human judgments (WMT QE), so the threshold is anchored to human‑aligned scores.
>
> As detailed in our response to Weakness 4, we further validated our data through rule-based sanity checks (verifying numerical preservation) and by reproducing the cross-lingual collapse on an existing translated GSM8K dataset from other paper [5].
>
> These results strongly demonstrate that our findings are robust and not driven by potential translation artifacts.
>
> [4] Guerreiro et al.xcomet: Transparent machine translation evaluation through fine-grained error detection.
> [5] Chen et al. Breaking Language Barriers in Multilingual Mathematical Reasoning: Insights and Observations.
>
> ---
>
> Your feedback has been instrumental in this process, and we sincerely extend our gratitude for your invaluable insights. Should you have any inquiries or require clarifications about our rebuttal, please don't hesitate to reach out. We are eager to address any concerns and elucidate potential ambiguities in greater depth.

---

> ### Author Response · Authors · 2025-11-27
> **Waiting for your feedback on our response**
>
> Dear reviewer o4RZ,
>
> We look forward to your feedback on our rebuttal. We greatly appreciate your engagement, which has helped us ensure that our detailed clarifications regarding novelty and experimental settings, as well as the additional ablation experiments, fully address the concerns raised. In light of these modifications, we respectfully request that you reconsider the evaluation of our paper. Within 6 days remaining in the discussion period, we stand ready to promptly address any remaining concerns or questions.
>
> Thank you for your feedback.
>
> Sincerely,
>
> Authors

---

> ### Author Response · Authors · 2025-12-03
> **Summary of Revisions**
>
> Dear reviewer o4RZ,
>
> We sincerely thank for your constructive feedback. In response, we have carefully revised and enhanced our manuscript by incorporating additional experiments and discussions during the rebuttal period:
>
> - **Novelty (W1).** We refined the list of contributions to more clearly highlight our work relative to prior work (**Section 1**).
>
> - **Generalization to Larger Models (W2).** We validated our findings on larger backbones, as reported in **Appendix E**.
>
> - **Expanded language coverage  (W3).** We added Japanese and Thai to our main experiments (**Table 1 & Section 3**) and clarified our rationale for selecting languages spanning high/mid/low-resource settings in **Section 3.1**.
>
> - **Translation Quality (W4, Q2).** We performed  an additional sanity check on the quality of the translated datasets, including numerical consistency verification (**Appendix B**).
>
> - **Qualitative Analysis (W5)**: We analyzed model behavior by comparing performance on the same questions in English and in the target language, which confirms that translated prompts are routed into the model's stronger English reasoning pathways (**Section 3.2, Appendix F**). We also conducted a detailed qualitative analysis of reasoning traces to categorize collapse patterns and scaffolding behaviors (**Appendix D**).
>
> We appreciate your time and effort in evaluating our work. Your comments have been instrumental in strengthening our manuscript.
>
> Sincerely,
>
> Authors

---

### Official Review · Reviewer_RNLe · 2025-11-02

**Soundness:** 3
**Presentation:** 3
**Contribution:** 2
**Rating:** 2
**Confidence:** 3

**Summary:**

This paper demonstrates that training with verifiable rewards and no incentive for the model to output tokens in a particular language results in the model outputting reasoning tokens in English. It also shows that efforts to mitigate this issue reduce accuracy.

**Strengths:**

1. Overall the explanation in the paper is clear, with nice illustrative figures.
2. The experimental results are reasonably extensive.
3. The paper is covering an important topic of making sure that multilingual models know how to reason in different languages.

**Weaknesses:**

1. The insight that LLMs don't reason consistently in the language of the original question is somewhat well known already, including in early reports from deepseek and openai where they observed that the reasoning chains resulting from RLVR are not consistently "good" English.
2. The methods to attempt to force the reasoning chains to be in the original language are not very effective, they do not demonstrate that it is possible to get good performance when reasoning in other languages.

Based on this, I'm not exactly sure how this paper would change practice in any way.

**Questions:**

What parts of the paper do you think are the ones that are going to be the most impactful to people who are training or using language models?

---

> ### Author Response · Authors · 2025-11-21
>
> Dear reviewer RNLe,
> We express our sincere gratitude for the invaluable feedback offered by the reviewer.
>
> ---
>
> **[Weakness 1] The insight that LLMs don't reason consistently in the language of the original question is somewhat well known already, including in early reports from deepseek and openai where they observed that the reasoning chains resulting from RLVR are not consistently "good" English.**
>
> We agree with the reviewer that prior technical reports from DeepSeek and others have already noted that RLVR-trained models often produce reasoning traces that are not consistently fluent or “good” English. Our goal is not to claim that such irregularities were previously unknown, but to make this phenomenon precise, measurable, and configurable in a multilingual setting. Concretely, we (i) coin and operationalize Cross-lingual Collapse as a joint pattern of rising task accuracy and systematic drift of CoT into the pre-training–dominant language, quantified via accuracy and target/English word-ratio metrics; (ii) show that this language switch reliably appears in standard, lab-scale RLVR setups; and (iii) analyze both the factors that amplify or attenuate the drift and the rollout-level mechanisms that drive it.
>
> While the DeepSeek [1] reports qualitatively mention language mixing and degraded readability, they do not provide a formal definition of the phenomenon, characterize when and why it appears under non-English prompts, or systematically examine how it can be mitigated. In contrast, we deliberately work in a controlled, reproducible setting to isolate Cross-lingual Collapse and study simple, intuitive interventions (language-consistency reward, decoding controls, multilingual GRPO) that disentangle its causes and reveal its persistent performance–fidelity trade-off. As we note in the paper, this trade-off ultimately stems from English-dominant pre-training priors and thus cannot be fully removed; however, making the failure mode explicit provides actionable insight for practitioners who need multilingual reasoning models with predictable and controllable language behavior.
>
> [1] Guo, Daya, et al. "Deepseek-r1: Incentivizing reasoning capability in llms via reinforcement learning." arXiv preprint arXiv:2501.12948 (2025).
>
> ---
>
> **[Weakness 2] The methods to attempt to force the reasoning chains to be in the original language are not very effective, they do not demonstrate that it is possible to get good performance when reasoning in other languages.**
>
> We appreciate the reviewer’s concern and agree that our proposed interventions do not fully eliminate cross-lingual collapse nor close the gap to English-level performance. This is, in fact, a central point of the paper. As stated in the Introduction, our goal is not to introduce a novel, highly engineered mitigation algorithm, but to systematically analyze cross-lingual collapse under the very mitigation strategies that are already commonly used in RLVR settings (and also adopted by the works the reviewer refers to): language-consistency rewards, decoding controls, and multilingual training mixtures.
>
> Our experiments show that these “standard” knobs behave in a consistent performance–language-fidelity trade-off regime: whenever we push the model toward higher target-language word ratio (via language-consistency reward, entropy reduction, or multilingual GRPO), we observe a measurable drop in accuracy, whereas letting the model freely seek reward leads to strong performance but severe drift into English. We view this negative result as an important empirical finding: it indicates that, given English-dominant pre-training priors, such collapse is a structural consequence of current RLVR objectives rather than something that can be removed by simply tuning known mitigations.
>
> In sum, our analysis shows that cross-lingual collapse is a structural consequence of current reward and exploration design.
>
> ---
>
> We thank the reviewer once again for the thoughtful and constructive feedback. We will revise the manuscript to more clearly articulate (i) the scope and intent of our analysis-centric framing, (ii) the implications of the observed performance–fidelity trade-off, and (iii) the limitations of existing mitigation strategies within RLVR. We appreciate the reviewer’s insights, and we will ensure that the revised version explicitly clarifies these points to improve readability and technical precision.

---

> > ### Comment · Reviewer_RNLe · 2025-11-21
> > **Thank You**
> >
> > Thank you for the response, I think it reinforces my original assessment and will stick with my rating.

---

> > > ### Author Response · Authors · 2025-11-21
> > >
> > > Thank you again for taking the time to review our work.
> > > To better understand your assessment in a constructive way, could you share which specific aspects of the submission most strongly contributed to the negative rating?
> > >
> > > If possible, we would be grateful for 1 or 2 actionable suggestions that you feel would meaningfully strengthen the paper.
> > >
> > > We genuinely value your perspective, and we are willing to run additional targeted experiments during the discussion period if your guidance can help us address the concerns more directly.

---

> ### Author Response · Authors · 2025-11-27
> **Further discussion before the deadline**
>
> Dear reviewer RNLe,
>
> We look forward to your feedback on our rebuttal. We believe that incorporating your constructive feedback regarding the scope and limitations of the prior work has strengthened the contribution of our paper. In light of these modifications, we respectfully request that you reconsider the evaluation of our paper. With 6 days remaining in the discussion period, we stand ready to promptly address any remaining concerns or questions.
>
> Thank you for your feedback.
>
> Sincerely,
>
> Authors

---

> ### Author Response · Authors · 2025-12-03
> **Summary of Revisions**
>
> Dear reviewer RNLe,
>
> We sincerely thank for your constructive feedback. In response, we have carefully revised and enhanced our manuscript by incorporating additional experiments and discussions during the rebuttal period:
>
> - **Novelty (W1, W2, Q1)**. We have refined the contribution bullet points in **Section 1** to more clearly highlight our contributions relative to prior work. We also expanded our experiments across languages and model scales to validate the generality of our findings (see **Table 1, Section 3.2, and Appendix E**).
>
> We appreciate your time and effort in evaluating our work. Your comments have been instrumental in strengthening our manuscript.
>
> Sincerely,
>
> Authors

---

### Official Review · Reviewer_JkaA · 2025-11-03

**Soundness:** 3
**Presentation:** 3
**Contribution:** 3
**Rating:** 6
**Confidence:** 4

**Summary:**

This work studies how verifiable-reward training (GRPO) on translated reasoning datasets causes language drift in multilingual LLMs. The authors show that as reasoning accuracy rises, chains of thought collapse into the model’s pretraining-dominant language (English)—a phenomenon they call Cross-lingual Collapse. They systematically measure, explain, and partially mitigate it via reward shaping, controlling generations, and multilingual training, revealing a persistent accuracy–fidelity trade-off that challenges multilingual reasoning alignment under RLVR.

**Strengths:**

- The authors identify a consistent phenomenon and analyze mechanisms that both exacerbate and mitigate the phenomenon in an easy-to-follow manner.
- The study is comprehensive across language choice, base model, and evaluation datasets.
- I think the results are interesting and highlights a scenario where the ‘unconditional reward seeking’ behavior of the RL optimization leads to undesirable effects.

**Weaknesses:**

- There are a few things unclear about the setup and main results in Table 1, namely the baseline that the authors are comparing to and the persistent non-zero EN WR; please see Questions.
- Some of the details in the authors’ implementation of their mitigation strategies require further clarification; please see Questions.

Minor:
- Some areas in the manuscript need `\citep` (eg. line 99, 162)
- Wording in sentence from lines 166-167
- Line 195: Accuracy -> accuracy
- Line 196: -34.0 -> +34.0
- Wording in sentences from lines 201-203
- Line 357: Ukrainia -> Ukrainian

**Questions:**

- I’m a bit unclear on the ‘non fine-tuned baseline’ that the authors compare to in Table 1; did the authors conduct a cold start SFT phase using translated versions of the datasets or does this start from the base model? How much does the gap get mitigated with more dedicated SFT on a diverse set of reasoning datasets (not just GSM8k, but adding eg. translated OpenMathInstruct2, etc).
- It seems that for most of the baselines, all of the EN WR is still quite high (~15-20%) - what are these tokens comprised of in the reasoning traces? It seems that it is possible to have near-zero EN WR as exemplified by Qwen’s 0.5 WR for Ukrainian. Including some example traces from the baseline models would be useful in this regard.
- How do the authors implement the language consistency reward? Is this done by a threshold for EN WR or an added bonus related to Target WR? How would an alternative strategy where authors discard rollouts containing English CoT continuations (but ensuring they sample enough rollouts to fill their batch) perform?
- If the authors compared “English-drifted” answers to regular completions from the original English versions of the training datasets, how similar or different are they?

---

> ### Author Response · Authors · 2025-11-21
>
> Dear Reviewer JkaA,
> We appreciate your constructive comments.
>
> ---
>
> **[Questions 1] I’m a bit unclear on the ‘non fine-tuned baseline’ that the authors compare to in Table 1; did the authors conduct a cold start SFT phase using translated versions of the datasets or does this start from the base model? How much does the gap get mitigated with more dedicated SFT on a diverse set of reasoning datasets (not just GSM8k, but adding eg. translated OpenMathInstruct2, etc).**
>
> Thank you for pointing out the ambiguity  of 'non finetuned baselies' in Table 1 and 6. In Tables 1 and 6, 'non fine-tuned baseline' refers to the instruct tuned models for each backbone (Qwen: Qwen-2.5-1.5B instruct, Llama: Llama-3.2-3B instruct, OLMo: OLmo2-Instrcut).  All of our finetuning starts from these instruction models. The umbers in
> parentheses indicate the change relative to those models. we will make clear this cpation to "off-the-shelf instruct baseline" in the paper.
>
>
> Regarding to SFT training with additional dataset (non-gsm8k dataset), we agree it is a valuable experiment. Unfortunately, we can not include this experiment due to limited GPU budget and rebuttal-period time (Nov 20 AOE).  We will report the experiment result untill the discussion period ( Dec 2, AOE). Based on our current analysis, such SFT should raise initial Target WR and delay drift, yet RLVR pressure can still yield the accuracy–fidelity trade‑off.
>
> ---
>
> **[Questions 2] It seems that for most of the baselines, all of the EN WR is still quite high (~15-20%) - what are these tokens comprised of in the reasoning traces? It seems that it is possible to have near-zero EN WR as exemplified by Qwen’s 0.5 WR for Ukrainian. Including some example traces from the baseline models would be useful in this regard.**
>
> Thank you for insightful comment of discussing non trivial english word ratio cases. We investigated why finetuned models still show non‑trivial English word ratio (EN WR ~15–20%) through carefully reviewing samples having relative high EN WR.
> First, finetuned models sometimes generate short English scaffolding or terminology inside chain-of-thought in the target lanugage. Second, Latin alphabet technical tokens remain in the reasoning, including algorithm or function names and unit abbreviations that are not localized. By contrast, cases with near‑zero EN WR do appear, such as Qwen on Ukrainian, where the model keeps both scaffolding and technical terms in the target language. We will add representative samples for both patterns to the revised manuscript.
>
> Below is the example of each Cases
>
> **Case 1 : Scaffolding or terminology inside chain-of-though**
>
> 1. **Initial Investment**: 카를로스는 레몬 나무를 심는데 90달러가 들었습니다.
>
> 2. **Annual Revenue**: 매년 7개의 레몬이 열리고, 각각 1.5달러에 팔 수 있습니다. 따라서 매년 수익은 7 * 1.5 = 10.5달러입니다.
>
> 3. **Annual Expenses**: 매년 3달러의 비용이 듭니다.
>
> 4. **Net Annual Profit**: 매년 수익에서 비용을 뺀 결과는 10.5 - 3 = 7.5달러입니다.
>
> 5. **Time to Break Even**: 초기 비용을 빼고, 매년 수익을 빼면, 카를로스가 수익을 내기 시작하려면 90 / 7.5 = 12년이 걸립니다.
>
> 최종 답: \boxed{12} 년
>
> **Case 2:  Using Latin alphabet**
>
> 为了找到从任意一点 A、B、C 或 D 出发，并访问其他三个点各一次的最短行程距离，我们可以考虑将问题转化为一个图论问题，其中每个点代表一个节点，每条边代表一个距离。我们需要找到从任意一个节点出发，经过其他三个节点，最后回到原点的最短路径。
>
> 根据题目给出的图，我们可以看到：
> - A 到 $B$ 的距离是 3
> - A 到 $C$ 的距离是 6
> - A 到 $D$ 的距离是 5
> - B 到 $C$ 的距离是 6
> - B 到 $D$ 的距离是 4
> - C 到 $D$ 的距离是 5
>
> 为了找到最短路径，我们可以使用图论中的最短路径算法，如Dijkstra算法。但是，由于这是一个简单的图，我们可以直接计算。
>
> 首先，我们从 A 出发，访问 B、C 和 D 各一次，然后回到 A。最短路径可以通过以下步骤计算：
> 1. 从 A 到 $B$ 的距离是 3。
> 2. 从 B 到 $C$ 的距离是 6。
> 3. 从 C 到 $D$ 的距离是 5。
> 4. 从 D 到 $A$ 的距离是 5。
>
> 因此，最短路径是 $A \rightarrow B \rightarrow C \rightarrow D \rightarrow A$，总距离是 $3 + 6 + 5 + 5 = 19$。
>
> 所以，最短的行程距离是 $\boxed{19}$。
>
> ---

---

> ### Author Response · Authors · 2025-11-21
>
> **[Questions 3] How do the authors implement the language consistency reward? Is this done by a threshold for EN WR or an added bonus related to Target WR? How would an alternative strategy where authors discard rollouts containing English CoT continuations (but ensuring they sample enough rollouts to fill their batch) perform?**
>
> Thank you for the question on cross-lingual consistency experiment setups.
> In our experiments, we implemented a dense additive language consistency reward following prior work [1], where the reward is calculated based on the ratio of words in the target language. To explore alternative design strategies, we also implemented a sparse reward, where the model obtain a reward of 0.5 if the generated response consists of at least 70% target language words. Moreover, to prevent reward hacking (e.g., generating extremely short responses in the target language), we set this reward to zero if the generated sequence length was fewer than 10 tokens.
>
> Consequently, both dense and sparse reward approaches successfully preserved language fidelity but underperformed vanilla GRPO in terms of accuracy. This result aligns with our observation that constraining the reasoning trace to a non‑English language blocks the "high‑yield shortcut" (i.e., English reasoning) and thus sacrifices part of the accuracy gain.
>
> Regarding the suggested experiment to discard rollouts containing English CoT continuations, we are currently implementing this strategy. Due to limited computational resources, we could not finalize the results immediately, but we will report these results as soon as possible within the discussion phase.
>
> (Note: Updated a rollout-level filtering result reflecting the comment after uploading initial response.)
>
> | Model |Language | GSM8K | TR(WR) | EN(WR) |
> |---------|--------|------|-----|----|
> |GRPO| UK|70.9| 0.3 | 96.8|
> |GRPO + lang loss (proportion)| UK| 62.5| 81.6| 16.4|
> |GRPO + lang loss (threshold: 0.7) | UK| 62.1| 85.0| 11.9|
> |GRPO + Rollout-level filtering | UK | 63.8 | 88.0 | 13.0 |
>
>
> [1] DeepSeek-AI. DeepSeek-R1: Incentivizing Reasoning Capability in LLMs via
> Reinforcement Learning
>
> ---
>
> **[Questions 4] If the authors compared “English-drifted” answers to regular completions from the original English versions of the training datasets, how similar or different are they?**
>
> Thank you for the helpful suggestion. In response to this question, we analyze the behavior of OLMo‑2 GRPO trained on GSM8K(KO). Interestingly, among the items the fine-tuned model answered correctly, 78.3% were also solved when run as regular English completions on English GSM8K. The remaining 21.7% were solved only under the Korean prompt. Overall, the gains observed under drift largely reflect the model routing its reasoning through a stronger English prior—consistent with our paper’s evidence that accuracy rises as English tokens dominate the CoT.
>
> Thank you for the insightful experiment suggestion. It will strengthen our observation and improve the understanding of our work. We will add this additional experiment to the revision during the rebuttal period.

---

> ### Author Response · Authors · 2025-11-27
> **Waiting for your feedback on our response**
>
> Dear reviewer JkaA,
>
> We look forward to your feedback on our rebuttal. We greatly appreciate your engagement, which has helped us ensure that our clarifications regarding the experiment settings and our additional analysis fully address the concerns raised. I
> In light of these modifications, we respectfully request that you reconsider the evaluation of our paper. Within 6 days remaining in the discussion period, we stand ready to promptly address any remaining concerns or questions.
>
> Thank you for your feedback.
>
> Sincerely,
>
> Authors.

---

> ### Author Response · Authors · 2025-12-03
> **Follow-up Experiments from our Previous Responses**
>
> ---
> **Update on Q1: Additional SFT Experiments**
>
>  As promised, we ran additional supervised fine-tuning (SFT) experiments using a more diverse Korean math corpus, combining GSM8K(KO) with the translated SimpleRL‑Zoo subset (“Base+Hard” curriculum). The new results are now reported in Section 3.3 and Appendix I (Table 10) of the revised manuscript.
>
> SFT(Base+Hard) improves accuracy over SFT(Base) while keeping Target WR very high on both GSM8K and MATH500. However, GRPO(Base+Hard) still achieves the highest accuracy, at the cost of severe language drift. These result also supports our observation of accuracy-fidelity trade off.
>
> ---
> **Update on Q3: Mitigation result using Rollout-level Filtering**
>
> We have updated the mitigation experiments to include the suggested algorithm that filters out English‑drifted rollout samples. This “rollout‑level filtering” method is now described and reported in Section 3.4 and Appendix H (Table 9) of the revised manuscript. Although this method effectively mitigates English drift, it still confirms our central finding: any strong constraint that enforces target‑language CoT reduces the accuracy gains obtained by unconstrained GRPO.
>
> ---

---

> ### Author Response · Authors · 2025-12-03
> **Summary of Revisions**
>
> Dear reviewer JkaA,
>
> We sincerely thank for your constructive feedback. In response, we have carefully revised and enhanced our manuscript by incorporating additional experiments and discussions during the rebuttal period:
>
> - **Clarification on Baselines and Additional SFT (Q1,W1).** We clarified how the baselines in **Table 1** are defined and added a comprehensive comparison with supervised fine-tuning (SFT) baselines, confirming that the performance–fidelity trade-off persists (**Appendix I**).
>
> - **Qualitative Analysis of English Tokens (Q2,W1).** We conducted a detailed qualitative analysis of reasoning traces to categorize the sources of non-zero English Word Ratio (EN WR) and included representative examples (**Section 3.2, Appendix D**).
>
> - **Mitigation Implementation and Additional Strategies (Q3,W2).** We detailed our language-consistency reward implementation and added experiments on the threshold reward and rollout-level filtering strategies suggested by the reviewer (**Section 3.4, Appendix H**).
>
> - **Cross-lingual Consistency Analysis (Q4).** We added a cross-lingual consistency analysis comparing performance on the same questions in English and in the target language,  which confirms that translated prompts are routed into the model's stronger English reasoning pathways (**Section 3.2, Appendix F**).
>
> - **Typo and formatting.**  We corrected citation-formatting inconsistencies and typos throughout the manuscript.
>
> We appreciate your time and effort in evaluating our work. Your comments have been instrumental in strengthening our manuscript.
>
> Sincerely,
>
> Authors

---

### Official Review · Reviewer_aLBm · 2025-11-06

**Soundness:** 2
**Presentation:** 2
**Contribution:** 3
**Rating:** 4
**Confidence:** 4

**Summary:**

This paper investigates a phenomenon termed Cross-lingual Collapse: when LLMs are trained with reinforcement learning using verifiable rewards (e.g., GRPO), their chain-of-thought reasoning in non-English target language prompts systematically drifts back toward the model’s pre-training dominant language (typically English), even when the prompts and training data are fully translated into the target language. This drift coincides with improved task accuracy, demonstrating a trade-off between performance and language fidelity. The work also evaluates several mitigation strategies (e.g., adding a language-consistency reward signal, reducing decoding entropy,  using multilingual training mixtures) and finds that while these approaches help maintain target-language fidelity, they also degrade reasoning accuracy to varying degrees.

**Strengths:**

- The paper provides a systematic identification and characterization of Cross-lingual Collapse, a previously underexplored failure mode in RLVR-based long chain-of-thought reasoning. The authors clearly illustrate how models trained with verifiable rewards tend to revert to their pre-training dominant language. This constitutes a meaningful contribution that advances our understanding of multilingual reasoning dynamics in large language models. The results suggesting that RLVR-based reasoning amplifies pre-training language priors also raise important concerns for multilingual reliability and interpretability.

- The study offers extensive empirical analysis and experiments with several plausible mitigation strategies, including language-consistency reward shaping, decoding-time constraints, and multilingual training mixtures. These interventions yield actionable insights into the trade-offs between reasoning performance and language fidelity, and provide valuable guidance for practitioners developing multilingual reasoning models. In addition, the paper is also well-written and easy to follow.

**Weaknesses:**

- The experiments examine only three non-English languages (Korean (KO), Ukrainian (UK), and Chinese (ZH)), which seems insufficient to support their broader claim of multilingual contexts and limit the generalizability of the presented collapse phenomenon. Also, the reason behind adopting these three languages lacks explanation (please correct me if I am wrong!).

- The training setup relies heavily on GSM8K, a widely used dataset that the backbone models may have already encountered during pre-training, potentially reinforcing existing language priors and introducing a confounding factor. Using a more recent dataset such as MATH for training, and evaluating with higher-quality human-curated MGSM [1] instead of machine-translated GSM8k, could greatly strengthen the validity of the empirical results.

- In the reported main experiments in Table 1, for the Ukrainian results on GSM8K, Qwen maintains a high target-language word ratio with no drift, whereas Llama exhibits a dramatic reduction in target-language word ratio (over an 80% decrease), despite Ukrainian being a low-resource language for both model contexts. This raises concerns that the extent of cross-lingual collapse may be model-dependent rather than a universal behavior. Conducting experiments on additional languages could further strengthen this conclusion (see the weakness noted above).

[1] Shi et al. "*Language models are multilingual chain-of-thought reasoners.*" ICLR 2023.

**Questions:**

- Some references are not consistently formatted throughout the paper in their use of \citet and \citep. For example, Marchisio et al. (2024) (Line 99), Guerreiro et al. (2024) (Line 155), and Lightman et al. (2024) (Line 162)

---

> ### Author Response · Authors · 2025-11-21
>
> Dear reviewer aLBm, We are grateful for the valuable feedback provided by the reviewer.
> In this response, we thoroughly address each of your comments and concerns.
>
> ---
>
> **[Weakness 1] The experiments examine only three non-English languages (Korean (KO), Ukrainian (UK), and Chinese (ZH)), which seems insufficient to support their broader claim of multilingual contexts and limit the generalizability of the presented collapse phenomenon. Also, the reason behind adopting these three languages lacks explanation.**
>
> We thank the reviewer for raising this point. We agree that limiting our main experiments to three non-English languages constrains the perceived breadth of our empirical coverage, and we will revise the paper to clearly state both our language-selection rationale and the intended scope of our claims.
>
> Our goal was not to assert that collapse universally occurs for all languages, but to study the phenomenon under controlled yet diverse linguistic conditions. The original three languages—Chinese (ZH), Korean (KO), and Ukrainian (UK)—were deliberately chosen to span (i) distinct scripts and (ii) a gradient of pre-training resource levels typical in modern LLMs (high-resource ZH, mid-resource KO, low-resource UK), while ensuring reliable translation and quality filtering. This design allowed us to examine how RLVR interacts with different pre-training linguistic priors rather than presenting only a single non-English case.
>
> As we emphasize in the paper, the severity of Cross-lingual Collapse depends jointly on the backbone’s pre-training distribution and the resource level of the target language. Languages well covered during pre-training tend to maintain high target-language fidelity under RLVR, whereas under-represented languages more readily drift toward the backbone’s dominant language (often English) as RLVR begins aggressively optimizing verifiable reward.
>
> To make this dependence more explicit, we expanded our study to two additional languages—Japanese (ja) and Thai (th)—covering both mid- and low-resource categories across OLMo2-1.5B, Llama-3-3B, and Qwen-1.5B. These supplementary experiments reveal the same pattern we report in the main paper:
>
> - English-centric backbones collapse strongly for under-represented languages, showing large accuracy gains accompanied by drastic drops in target-language word ratio and sharp increases in English word ratio.
>
> - More multilingual backbones maintain target-language reasoning for well-covered languages but still trace a similar performance–fidelity trade-off.
>
> We will include a summary table in the supplement to make the model- and language-dependent nature of Cross-lingual Collapse explicit.
>
> | Model |Language | GSM8K | TR(WR) | EN(WR) |
> |---------|--------|------|-----|----|
> |OLMO2-1.5B| ja |52.61 (+45.4)  | 2.3(-83.9) | 97.0 (+80.1)|
> |OLMO2-1.5B| th | 29.7(+27.2) | 1.4(-80.3)|  90.5( 81.0) |
> |Qwen-1.5B| ja | 43.1 (+7.5)| 95.8(-0.7)| 2.5 (+1.0)|
> |Qwen-1.5B | th| 68.3 (+31.7)|8.3 (-78.9) |89.9 (+80.5)|
>
> ---

---

> ### Author Response · Authors · 2025-11-21
>
> **[Weakness 2] The training setup relies heavily on GSM8K, a widely used dataset that the backbone models may have already encountered during pre-training, potentially reinforcing existing language priors and introducing a confounding factor. Using a more recent dataset such as MATH for training, and evaluating with higher-quality human-curated MGSM instead of machine-translated GSM8k, could greatly strengthen the validity of the empirical results.**
>
> We agree that dataset choice and evaluation quality are important, but we do not believe our conclusions hinge on GSM8K or on using machine‑translated data.
>
> GSM8K and possible pre‑training exposure. GSM8K is the standard math‑reasoning corpus. Even if our backbones partially saw GSM8K during pre‑training, this cannot explain the phenomenon we study: we fine‑tune only on non‑English translations of GSM8K, yet under GRPO the CoT systematically drifts back to English as accuracy improves, and this pattern appears consistently across three languages and multiple models.
>
> Using MATH / harder datasets. We agree that training on more challenging data is valuable. Our setup already partially follows this suggestion by mixing in SimpleRL‑Zoo, which contains harder MATH‑like problems. As shown in Table 2, adding this harder data in Korean initiates collapse: accuracy increases, while the Korean word ratio sharply drops and English tokens dominate. This indicates that cross‑lingual collapse is not specific to GSM8K but persists under a harder curriculum. In the revision we will clarify this connection and describe SimpleRL‑Zoo more explicitly as a higher‑difficulty complement to GSM8K.
>
> MGSM vs. our translated GSM8K. For evaluation, MGSM [1] provides only a small number of manually translated GSM8K items in a limited set of languages, and in particular does not include Korean or Ukrainian. We therefore construct larger multilingual versions of GSM8K and SimpleRL‑Zoo using GPT‑4 translations with automatic quality filtering (see §3.1 and App. B) to obtain sufficient coverage and scale.
>
> [1] Shi et al. Language Models are Multilingual Chain-of-Thought Reasoners
>
> ---
>
> **[Weakness 3] In the reported main experiments in Table 1, for the Ukrainian results on GSM8K, Qwen maintains a high target-language word ratio with no drift, whereas Llama exhibits a dramatic reduction in target-language word ratio (over an 80% decrease), despite Ukrainian being a low-resource language for both model contexts. This raises concerns that the extent of cross-lingual collapse may be model-dependent rather than a universal behavior. Conducting experiments on additional languages could further strengthen this conclusion.**
>
> We thank the reviewer for this observation. We agree that the extent of cross-lingual collapse is model-dependent, and this is consistent with how we frame the phenomenon in the paper: RLVR amplifies the linguistic priors induced by the pre-training corpus, so different backbones will exhibit different levels of collapse depending on how well the target language is represented in their pre-training mix.
>
> In particular, as discussed in our experimental setup and Section 4.1, we explicitly distinguish English‑dominant vs more multilingual backbones. Qwen‑2.5 is closer to the latter category, and its stronger multilingual prior means that, for Ukrainian GSM8K, RLVR does not find a strong incentive to reroute the reasoning trace through English—hence the near‑100% Ukrainian WR in Table 1. By contrast, Llama and especially OLMo have weaker Ukrainian priors; under the same RLVR objective they more readily exploit English trajectories, leading to the dramatic drop in Ukrainian WR that we label as cross-lingual collapse.
>
> Thus, we do not claim that every model–language pair will collapse in the same way; rather, our claim is that RLVR introduces a general pressure to exploit high‑resource, pre‑training‑dominant languages, whose manifestation depends on the backbone’s prior. This is also why, for the same Qwen backbone, we do observe collapse under harder curricula and in other languages, supporting the view that the mechanism is shared while its severity is model- and language-dependent.
>
> We agree that adding more languages would further strengthen the typological coverage. Within our compute and space budget we intentionally chose three languages spanning different scripts and resource levels (ZH/KO/UK) to illustrate this gradient, and we will clarify in the revision that extending to a broader set of languages is a natural direction for future work.
>
> ---
>
> **[Question 1] Some references are not consistently formatted throughout the paper in their use of \citet and \citep.**
>
> We appreciate you pointing out the inconsistencies in the reference format. We will carefully proofread all citations to ensure consistency and will upload the revised manuscript during the discussion period.

---

> > ### Author Response · Authors · 2025-11-27
> > **Waiting for your feedback on our response**
> >
> > Dear reviewer aLBm,
> >
> > We look forward to your feedback on our rebuttal. We greatly appreciate your engagement, which has helped us ensure that our clarifications regarding the structural limitations and our additional experiments fully address the concerns raised. In light of these modifications, we respectfully request that you reconsider the evaluation of our paper. With 6 days remaining in the discussion period, we stand ready to promptly address any remaining concerns or questions.
> >
> > Thank you for your feedback.
> >
> > Sincerely,
> >
> > Authors

---

> ### Author Response · Authors · 2025-12-03
> **Summary of Revisions**
>
> Dear reviewer aLBm,
>
> We sincerely thank for your constructive feedback. In response, we have carefully revised and enhanced our manuscript by incorporating additional experiments and discussions during the rebuttal period:
>
> - **Expanded language coverage (W1, W3).** We added Japanese and Thai to our main experiments (**Table 1 & Section 3**) and clarified our rationale for selecting languages spanning high/mid/low-resource settings in **Section 3.1**.
>
> - **Generalization to larger models (W3).** We validated our findings using larger backbones, as reported in **Appendix E**.
>
> - **Clarification of harder baselines (W2).**  We refined **Sections 3.1 and 3.3** to clarify our use of the more challenging curriculum dataset, thereby demonstrating that collapse can be driven by task difficulty rather than dataset memorization.
>
> - **Formatting (Q1).**  We corrected citation-formatting inconsistencies and typos throughout the manuscript.
>
> We appreciate your time and effort in evaluating our work. Your comments have been instrumental in strengthening our manuscript.
>
> Sincerely,
>
> Authors

---

### Author Response · Authors · 2025-11-25
**Common Response**

We sincerely appreciate your thoughtful feedback and the time you spent reviewing our work.

As reviewers highlighted, our work identifies and systematically analyzes 'cross-lingual collapse,' an underexplored failure case in RLVR-based multilingual CoT (Reviewers aLBm, JkaA, and RNLe). We demonstrate strong empirical evidence (all reviewers) showing how optimization for rewards amplifies English-dominant priors at the cost of target language fidelity (Reviewer aLBm, JkaA). These results, supported by our mitigation studies, provide valuable insights for the multilingual research community (Reviewer aLBm).

In this constructive rebuttal, we would like to clarify two points: 1) Regarding  concerns of novelty (Reviewers o4RZ, RNLe), we emphasize that our contribution is an analysis‑centric study that formally defines and operationalizes cross‑lingual collapse in multilingual RLVR. This goes beyond prior qualitative observations by providing a precise, reproducible characterization of the phenomenon and its limits. 2) To address the concern of generalizability of our findings (Reviewers aLBm, o4RZ), we expanded our experiments to provide additional evidence that the observed collapse pattern persists across model scales and languages, thereby strengthening the empirical support for our claims without changing the conclusions.

We hope this summary helps to understand our contribution and the additional clarifications provided in the rebuttal.



Best regards,

Authors

---

### Meta-Review · Area_Chair_qhDU · 2026-01-04

**Summary:**

The reviewers primarily raise the following concerns:

1. Limited languages covered in the experiments (aLBm, o4RZ). The initial draft only included three languages with synthetic (machine-translated) data for evaluation.

2. The paper did not offer an effective solution to the claimed problem, i.e., inconsistency when reasoning in different languages (RNLe).

3. Reviewers also raised a few concerns about presentation and clarity.

**Reviewer Concerns:**

Concern 1 has been somewhat addressed by adding a few more languages, but the authors insisted on using machine-translated data, which raises concerns about the quality of the ground truth. Concern 3 has been addressed in the rebuttal.

I'm in agreement with Reviewer RNLe---basically it's unclear to me how the proposed methods will change the practice in multilingual reasoning.

**Reviewer Scores:**

Reviewer RNLe has been explicit in keeping their score. I don't think other reviewers will change the score: JkaA has already been positive, and this paper doesn't deserve an 8; for the other two reviewers, the authors' response does not effective resolve the concern regarding evaluation reliability.

---

### Decision · Program_Chairs · 2026-01-26

Reject